# Restoring Pruned Large Language Models via Lost Component Compensation

**Zijian Feng**[1]    **Hanzhang Zhou**[1]    **Zixiao Zhu**[1]    **Tianjiao Li**[1]    **Jia Jim Deryl Chua**[2]
**Lee Onn Mak**[2]    **Gee Wah Ng**[2]    **Kezhi Mao**[1,*]

[1]School of Electrical and Electronic Engineering, Nanyang Technological University, Singapore
[2]Home Team Science and Technology Agency (HTX), Singapore
{feng0119, hanzhang001, zixiao001}@e.ntu.edu.sg
{tianjiao.li, ekzmao}@ntu.edu.sg
{deryl_chua, mak_lee_onn, ng_gee_wah}@htx.gov.sg

## Abstract

Pruning is a widely used technique to reduce the size and inference cost of large language models (LLMs), but it often causes performance degradation. To mitigate this, existing restoration methods typically employ parameter-efficient fine-tuning (PEFT), such as LoRA, to recover the pruned model's performance. However, most PEFT methods are designed for dense models and overlook the distinct properties of pruned models, often resulting in suboptimal recovery. In this work, we propose a targeted restoration strategy for pruned models that restores performance while preserving their low cost and high efficiency. We observe that pruning-induced information loss is reflected in attention activations, and selectively reintroducing components of this information can significantly recover model performance. Based on this insight, we introduce RestoreLCC (Restoring Pruned LLMs via Lost Component Compensation), a plug-and-play method that contrastively probes critical attention heads via activation editing, extracts lost components from activation differences, and finally injects them back into the corresponding pruned heads for compensation and recovery. RestoreLCC is compatible with structured, semi-structured, and unstructured pruning schemes. Extensive experiments demonstrate that RestoreLCC consistently outperforms state-of-the-art baselines in both general and task-specific performance recovery, without compromising the sparsity or inference efficiency of pruned models [2].

## 1 Introduction

Large language models (LLMs) have achieved remarkable success in various natural language processing (NLP) tasks like commonsense reasoning, math problem solving, and text completion [1, 2, 3]. However, their large parameter counts demand substantial computational resources for deployment and inference. To democratize LLMs, pruning has emerged as a key technique to reduce model size and accelerate inference [4]. Pruning typically involves two steps: weight pruning and performance restoration. Weight pruning methods can be roughly categorized into structured pruning, such as LLM-Pruner [5] and SlimGPT [6], semi-structured pruning, and unstructured pruning, such as SparseGPT [7] and Wanda [4]. These methods estimate the importance of parameters and zero out unimportant ones to reduce model size. Performance restoration is a critical step in mitigating the

---

*Corresponding author.
[2]Code: https://github.com/zijian678/restorelcc/

39th Conference on Neural Information Processing Systems (NeurIPS 2025).

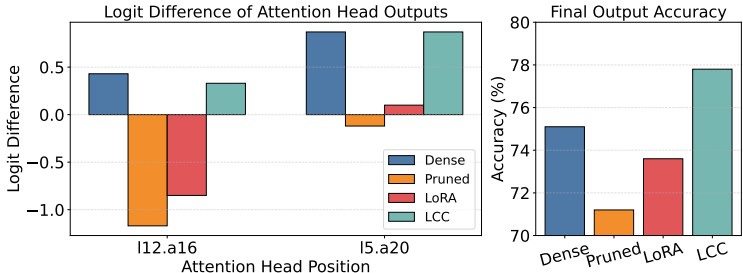

Figure 1: These figures compare the performance of the original **dense** model, the **pruned** model, and two recovery methods applied to the pruned model: **LoRA** and **Lost Component Compensation (LCC, ours)**, which directly adds back components lost due to pruning. The used dataset is BoolQ, and please see § 3 for experimental details. The left figure shows the logit difference (correct minus incorrect) of attention heads with/without recovery, and the right figure shows the accuracies of final model outputs for each method. *l[·]* and *a[·]* indicate the layer index and the head index, respectively. For instance, l12.a16 means the 16-th head at the 12-th layer.

performance degradation caused by weight pruning, aiming to recover model capability by adjusting weights through language modeling or instruction tuning datasets [8, 9, 10].

As full fine-tuning (FT) still requires substantial computational resources, parameter-efficient fine-tuning (PEFT) has become the mainstream approach for restoring pruned models. Recent pruning methods, including LLM-Pruner, SparseGPT, Wanda, and SlimGPT, all adopt LoRA [11] to recover performance. Although seemingly straightforward, applying existing PEFT methods, such as LoRA-based approaches [11, 12, 13] and representation engineering [14, 15, 16], to pruned LLMs raises concerns.

These PEFT methods are originally designed for dense models, where they adapt LLMs to downstream tasks by training a small subset of parameters. When applied to pruned models, they often overlook pruning-specific characteristics, such as the need to account for lost information, leading to inefficient parameter search and suboptimal restoration. As shown in Figure 1, attention heads recovered using LoRA achieve only limited improvement in logit predictions, resulting in suboptimal performance on the final task accuracy. In contrast, directly reintroducing the pruned components into the pruned attention heads (**LCC**) significantly improves both the outputs of the attention heads and the overall model accuracy. Built on this insight, we propose RestoreLCC (Restoring Pruned LLMs via Lost Component Compensation), which explicitly reconstructs critical information lost during pruning to bridge the performance gap between pruned and dense models.

Specifically, our study in § 3 reveals that important information removed by pruning can be captured in attention head activations, and reintroducing components of this lost information can substantially restore the performance of the pruned model. However, this insight also presents challenges, such as determining which attention heads to select and how to estimate component vectors that encode critical information for compensation. To overcome these challenges, RestoreLCC incorporates two main mechanisms: (1) **contrastive probing**, a general approach that leverages activation editing to construct contrastive sample pairs and probes a subset of attention heads critical for performance recovery; and (2) **lost component compensation** (LCC), which retrieves pruning-induced lost information from these key heads. This information is decomposed into components, represented as vectors that capture the lost information directions. We optimize their magnitudes to enable targeted restoration along these directions, aggregate them into a single informative component, and finally inject it into the pruned model to recover performance. Unlike existing PEFT methods that restore pruned LLMs in an unguided manner, RestoreLCC explicitly compensates for key components lost during pruning, offering a targeted and effective restoration strategy.

We empirically evaluate RestoreLCC against other performance restoration methods across all three types of pruned models (e.g., structured, semi-structured, and unstructured), on both general recovery and task-specific settings across a wide range of LLMs of different sizes. Our results show that RestoreLCC significantly improves pruned model performance under general recovery settings while maintaining similar inference speed and sparsity ratios, outperforming existing PEFT methods. Furthermore, under task-specific recovery settings, RestoreLCC successfully recovers task-specific information and enables higher pruning ratios, where other PEFT methods often fail.

**Contributions.** Our main contributions are: (1) We observe that pruning-induced information loss is reflected in attention activations, and that selectively restoring key components can significantly recover model performance (§ 3); (2) We propose RestoreLCC, a method that learns the magnitudes of important component directions lost during pruning and reintroduces them to restore pruned models effectively (§ 4); (3) Extensive experiments across various LLMs and pruning schemes demonstrate that RestoreLCC consistently outperforms existing restoration baselines (§ 5).

## 2 Related work

**LLM Pruning.** Pruning reduces model size and speeds up inference by removing less important weights. Unstructured pruning removes individual weights irrespective of position, as in SparseGPT [7], which uses approximate Hessian-based reconstruction, and Wanda [4], which ranks weights by magnitude and activation norms. DSOT [17] and ALPS [18] further refine sparsity via optimization. Semi-structured pruning enforces patterns like N:M sparsity [19], and many unstructured methods adapt to this format [4, 7, 18]. Structured pruning removes entire blocks (e.g., rows or columns) to enhance hardware efficiency. Recent methods include LLM-Pruner [5], Compresso [20], LoRAPrune [21], and SlimGPT [6].

**Performance Restoration and PEFT.** Pruning methods such as LLM-Pruner [5], LoRAPrune [21], SparseGPT [7], Wanda [4], and SlimGPT [6] consistently benefit from a performance restoration step, typically involving fine-tuning on language modeling (e.g., C4 [8], WikiText [9]) or instruction-tuning datasets (e.g., Alpaca [10]). While full fine-tuning (FT) is effective, it remains computationally expensive, often requiring days on large GPU clusters. Parameter-efficient fine-tuning (PEFT) offers a cheaper alternative by updating a small subset of parameters. PEFT techniques include adapter-based [22, 23, 24], prompt-based [25, 26], LoRA-based (e.g., LoRA [11], AFLoRA [27], VERA [12], DoRA [13]), and representation-engineering methods (e.g., RED [14], ReFT [15], LoFit [16]). Among them, LoRA is the most widely adopted in modern pruning pipelines. In addition to PEFT methods, FLAP [28] proposes a bias compensation strategy to reduce the pruning loss. EoRA [29] provides a fine-tuning-free approach to recover pruned models by searching low-rank spaces in a task-specific eigenspace to minimize compression loss.

## 3 Insight: injecting pruned components back effectively restores models

**Preliminaries.** We begin by clarifying several key terms and the scope of this study. Current LLM pruning methods primarily target the weight matrices of both attention and feed-forward (FFN) modules. For example, Wanda removes 50% of the weights in each matrix within both the attention and FFN modules (setting them to zero), resulting in an overall sparsity of 50%. Accordingly, in our work, **all weight matrices within the LLM are pruned, and we focus on restoring the pruned model by compensating through attention heads**. An *activation* refers to the output of a specific module within the Transformer architecture. A *pruned activation* is the output produced by a pruned module. Although its dimensionality remains identical to that of the original activation, it typically contains less information because the underlying weight matrices have been pruned. Importantly, the activations themselves are not pruned—only the associated weight matrices are. *Sparsity* denotes the proportion of weight parameters that have been pruned.

Pruning inherently leads to information loss, which becomes more pronounced at higher pruning ratios. In this section, we demonstrate that the pruned activations contain critical information, including discriminative components essential for downstream NLP tasks. By reintroducing the components, the performance of the pruned model can be substantially restored.

Recent studies have shown that different attention heads specialize in distinct functions when performing NLP tasks [30, 31, 32]. Motivated by this, we investigate how pruning-induced loss of head activations affects the model's retained functional capacity and performance. Let $\mathrm{MultiHead}$ denote the multi-head attention output in a Transformer block with $H$ attention heads. The output can be expressed as:

$$\mathrm{MultiHead}\left(x^l\right) = \mathrm{concat}\left(z^{(l,0)}, \ldots, z^{(l,H-1)}\right) W^O, \tag{1}$$

where $x^l$ is the input to the $l$-th layer, $z^{(l,h)}$ is the output of the $h$-th head, and $W^O$ is a shared output projection matrix. Denote by $z_d^{(l,h)}$ and $z_p^{(l,h)}$ the activations of the $h$-th head in the dense and pruned

models, respectively, for the same input. The lost activation can be computed as Eq. 2. Our objective is to analyze whether $\delta z^{(l,h)}$ carries critical information that could aid in model recovery.

$$\delta z_{d-p}^{(l,h)} = z_d^{(l,h)} - z_p^{(l,h)}. \tag{2}$$

Given $N$ samples, we denote the activation loss matrix for all samples as $\Delta \mathbf{Z}^{(l,h)} = [\delta z_{0,d-p}^{(l,i)}; \ldots; \delta z_{N-1,d-p}^{(l,h)}] \in \mathbb{R}^{N \times d_h}$, where $d_h$ is the dimensionality of the head activation. To better characterize the structure of the lost information, we apply singular value decomposition (SVD) to $\Delta \mathbf{Z}^{(l,h)}$, as shown in Eq. 3, which decomposes the activation loss matrix into a set of orthogonal latent components. This formulation enables us to identify the dominant directions of lost activation information and quantify their impact on each sample.

$$\Delta \mathbf{Z}^{(l,h)} = \mathbf{U}^{(l,h)} \Sigma^{l,h)} \mathbf{V}^{(l,h)\top} = \sum_{i=1}^{d_h} \sigma_i \, u_i^{(l,h)} \, v_i^{(l,h)} \approx \sum_{i=1}^{K} \sigma_i u_i^{(l,h)} v_i^{(l,h)}. \tag{3}$$

For the output $z_p^{(l,h)}$ of a pruned attention head, the lost principal components can be approximated by

$$c^{(l,h)} = \sum_{i=1}^{K} \alpha_i^{(l,h)} v_i^{(l,h)}, \tag{4}$$

where $c^{(l,h)} \in \mathbb{R}^{d_h}$, and $\alpha_i^{(l,h)}$ denotes the average of $\sigma_i u_i^{(l,h)}$, representing the mean projection coefficients across samples. Finally, the activation output of the pruned attention head $z_p^{(l,h)}$ can be compensated and recovered by injecting the estimated lost components $c^{(l,h)}$ back:

$$z_c^{(l,h)} = z_p^{(l,h)} + c^{(l,h)}, \tag{5}$$

where we omit the sample index $i$ in $z_{i,c}^{(l,h)}$ and $z_{i,p}^{(l,h)}$ for simplicity. To evaluate the predictive behavior of an attention head's activation, we follow the theories of LogitLens [33] and prior work on interpreting LLMs in embedding space [34]. Specifically, we project the activation into the embedding space and compute its prediction over the vocabulary space:

$$p_{z_c}^{(l,h)} = \text{LM\_Head} \left[ \phi \left( z_c^{(l,h)} W^{O,h} \right) \right], \tag{6}$$

where LM_Head denotes the LLM's prediction head, $\phi(\cdot)$ is the layer normalization function, $W^{O,h}$ is the output projection matrix in $W^O$ corresponding to the $h$-th head, $p_{z_c}^{(l,h)} \in \mathbb{R}^{|\mathcal{V}|}$, $\mathcal{V}$ represents the model's vocabulary, and $|\mathcal{V}|$ is the vocabulary size. Following IOI [35], we compute the **logit difference**, defined as the logit assigned to the correct token minus the logit assigned to the wrong token. This difference directly reflects the model's confidence and faithfulness in predicting the correct token over an incorrect alternative.

**Empirical Study.** To evaluate whether reintroducing lost components can aid in performance recovery, we conduct an empirical study using BoolQ [36], a widely used commonsense reasoning dataset. We adopt LLaMA-7B [2] as the dense backbone model and apply Wanda [4], a state-of-the-art pruning method, to prune the model to 50% sparsity. We randomly sample 1,000 examples from BoolQ, where both the dense and pruned models are tasked with answering "yes" or "no" questions. We extract the attention head activations from the pruned model and reconstruct the compensated activations using the main components, as described in Eqs. 2-5. To quantify model confidence, we compute the **logit difference** $\lambda$ using:

$$\lambda = p_{z_c}^{(l,h)}[\text{yes}] - p_{z_c}^{(l,h)}[\text{no}] \quad \text{or} \quad p_{z_c}^{(l,h)}[\text{no}] - p_{z_c}^{(l,h)}[\text{yes}],$$

We also report final **accuracy**, i.e., the predictions from the last layer, to assess the effect of component compensation on the model's output.

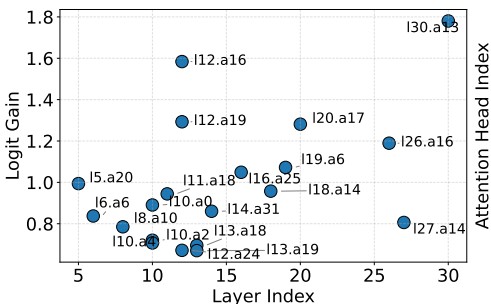
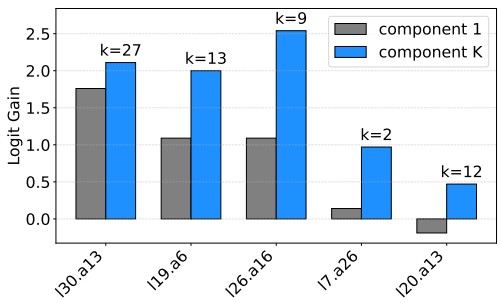

Figure 2: Logit gain of different attention heads recovered with principal components.

Figure 3: Logit gain of attention heads recovered with different components.

**Finding 1. Reintroducing lost components to selected pruned attention heads can significantly restore model performance.** As shown in Figure 1, we use the top-10 ($K = 10$ in Eq. 4) lost components to reconstruct the pruned activations. The left figure demonstrates that this compensation substantially recovers the pruned activations. Specifically, the logit difference is restored to a level comparable to that of the original dense model. Furthermore, compensating these attention heads also leads to a notable improvement in the model's final output accuracy.

**Finding 2. Different attention heads vary in importance and exhibit distinct recovery behaviors.** We manually select and examine the outputs of several attention heads, along with their restoration using the top-10 lost components. To quantify the direct effect of compensation, we calculate the **logit gains** as $\delta\lambda = \lambda_{\mathrm{recovered}} - \lambda_{\mathrm{pruned}}$, which directly captures the improvement brought by reintroducing the lost components. Figure 2 shows the recovered logit gains for these heads. The results reveal that attention heads respond differently to the compensation process, indicating that not all can be effectively restored using the lost principal components.

**Finding 3. Discriminative information may reside in minor components rather than in the principal ones.** Figure 3 illustrates the recovery performance using either the top principal component or a selected minor component. Notably, since the coefficients of minor components are extremely small, we scale them by a factor of 1000 for visualization and evaluation. Surprisingly, incorporating certain minor components can lead to substantially better restoration performance compared to using the leading principal component.

**Extension to FFN**. Each Transformer layer comprises a multi-head attention (MHA) module and a feed-forward network (FFN) module. In the above analysis, we focus on compensating for the MHA rather than the FFN. To further justify this design choice, that is, using MHA instead of FFN, we provide additional theoretical and empirical analyses in Appendix A.

To summarize, the above analysis highlights not only the strong potential of leveraging lost components for performance restoration, but also several key challenges: (1) How to select pruned attention heads that are important for recovery? (2) How to determine the positions and coefficients of key components? and (3) How to apply the findings to more general tasks beyond BoolQ?

## 4    RestoreLCC: restoring pruned LLMs via lost component compensation

To address these challenges and enable universal performance restoration for pruned LLMs, we propose RestoreLCC, shown as Figure 4, which consists of two key mechanisms: (1) contrastive probing, a general method that uses activation editing to create contrastive sample pairs and identify critical attention heads; and (2) lost component compensation (LCC), which optimizes the magnitudes of directional components for the lost information. The optimized components are then injected back into the model to restore performance.

### 4.1    Contrastive probing

**Contrastive Sample Construction.** We construct contrastive samples for general NLP tasks to support activation editing and probing to localize important heads. Given a dataset, either task-

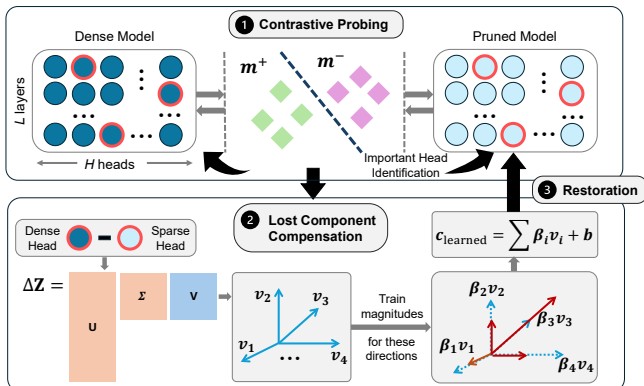

Figure 4: Overview of the RestoreLCC framework. It integrates two key modules, (1) contrastive probing and (2) lost component compensation (LCC), to restore the performance of pruned LLMs.

specific (e.g., BoolQ) or general (e.g., the Alpaca dataset), every sample is organized as $(q, r^+)$, where $q$ is the question and $r^+$ is the correct (positive) response. All responses are collected into a set $[r_0^+, \ldots, r_{N-1}^+]$. We use a sentence encoder, such as MiniLM-L6 [37], to encode all responses. For each sample, we select the most similar response (excluding the correct one) based on cosine similarity as the negative response $r^-$. This process converts each sample into a contrastive tuple $(q, r^+, r^-)$. Note that this method is universal and can be applied to any dataset. We provide examples of constructed samples from the BoolQ and Alpaca datasets in Appendix B.

**Activation Editing.** Recent studies suggest that high-dimensional activations in LLMs are approximately orthogonal with high probability [38]. An activation can be guided toward a desired generation space by adding a steering vector or a task-specific function vector [39, 32]. For pruned LLMs, we assume the lost principal components can be reintroduced to recover the activation and restore the correct response. Formally: recovered question activation $\approx$ pruned question activation $+$ lost important component. Based on this, the recovered activation for a question is defined as

$$z_c^q = z_p^q + c^q, \tag{7}$$

where we omit the layer and head indices $(l, h)$ for simplicity. Here, $q$ indicates the activation is taken from the last token of the question, and the other symbols follow the notation in Eq. 5.

**Attention Head Probing.** Ideally, the recovered activation $z_c^q$ should contain sufficient information to generate the correct response. It should be consistent with the activation of the complete and correct sample, denoted as $z_d^{q+r^+}$, which is obtained from the last token of the full sequence (including both the question $q$ and the positive response $r^+$) in the original dense model. Additionally, it should be contrastive with respect to the negative sequence activation $z_d^{q+r^-}$. However, some attention heads may be inactive, and certain components may contribute little to performance restoration. In such cases, the recovered activation $z_c^q$ may not adequately distinguish between $z_d^{q+r^+}$ and $z_d^{q+r^-}$.

In this way, identifying important attention heads can be formulated as a natural language inference (NLI) task. If an attention head is important and the corresponding component is useful, then the recovered activation $z_c^q$ should entail the correct sequence activation $z_d^{q+r^+}$ and contradict the negative sequence activation $z_d^{q+r^-}$. Specifically, activation pairs are constructed as $m^+ = [z_c^q, z_d^{q+r^+}]$ with label 1 (entailment) and $m^- = [z_c^q, z_d^{q+r^-}]$ with label 0 (contradiction). A probing classifier, composed of a linear layer followed by a sigmoid activation, is trained to assess the discriminative power of each recovered head activation. The importance of attention heads is then ranked based on the accuracy of their corresponding probing classifiers.

### 4.2 Lost component compensation

We now have a list of important attention heads along with their corresponding lost components $v_i$ based on Eq. 3. The goal is to use these components to approximate the missing information and recover the performance of pruned LLMs.

**Lost Component Estimation.** As observed in § 3, the usefulness of a component does not necessarily correlate with its coefficient. Minor components may also carry critical information and enable more effective recovery of pruned LLMs. To address this, we consider all possible components. Since the vectors $v_i$ are orthogonal and unit-normalized, we treat each $v_i$ as a potential direction of lost information. We keep these directions fixed and learn a scalar magnitude for each, representing its importance. Formally, for an attention head, we model its lost component containing important information as:

$$c_{\text{learned}} = \sum_{i=1}^{d_h} \beta_i v_i + b. \tag{8}$$

Here, $v_i$ is obtained from Eq. 3 and remains fixed during training, while $\beta_i$ is a trainable scalar indicating the importance of each direction. The term $b \in \mathbb{R}^{d_h}$ is a trainable bias vector for the attention head, acting as a hedging vector to provide flexibility in cases where important information lies outside the span of the predefined directions.

**Component Compensation.** For each pruned attention head, its output activation is recovered as $\tilde{z}_p = z_p + c_{\text{learned}}$. It is worth noting that the final learned component $c_{\text{learned}}$ is a constant bias vector, capturing important information that was lost for all samples as a result of pruning. The recovered activation $\tilde{z}_p$ is then passed to the subsequent computations, such as those described in Eq. 1, allowing the model to better approximate the behavior of the original dense LLM.

**Extension of RestoreLCC to FFNs.** It is worth noting that RestoreLCC can be seamlessly extended to FFN modules. Our objective is to restore pruned models in which all weight matrices—both in the attention and FFN modules—have been pruned. We provide further theoretical and empirical analyses in Appendix A to justify why we apply RestoreLCC to attention heads instead of FFNs.

### 4.3  Overhead analysis of sparsity and inference speed

LLM pruning aims to produce a sparse model, while performance restoration seeks to close the performance gap between the pruned and dense models. It is important to ensure that performance restoration improves the pruned model's accuracy without compromising its sparsity or inference efficiency.

**Sparsity Analysis.** Each compensated attention head introduces a learned vector $c_{\text{learned}} \in \mathbb{R}^{d_h}$. In the worst case, where all heads in a layer are compensated, the total number of additional parameters introduced across the projection matrices (`q_proj`, `k_proj`, `v_proj`, `o_proj`) within the multi-head attention mechanism is:

$$\text{Parameter Overhead} = \frac{2d_l}{4d_l^2} = \frac{1}{2d_l},$$

where $d_l$ is the hidden size of the layer and $d_h = d_l/H$. Since $d_l$ is typically larger than 1000, the increase in parameters is less than 0.05%, which has minimal impact on sparsity.

**Inference Speed.** To maintain inference efficiency, the learned vector $c_{\text{learned}}$ is absorbed as the constant bias vector in the multi-head attention block. This modification introduces almost no additional computation during inference and preserves the speed of the pruned model. Therefore, RestoreLCC effectively preserves the pruned model's sparsity and inference speed while recovering its performance. Additional empirical evidence is provided in Appendix D.

## 5  Experiments

### 5.1  Experimental settings

**Metrics.** To evaluate the restoration effectiveness of RestoreLCC, we experiment with representative pruned LLMs from diverse pruning strategies. Specifically, we use Wanda [4] for unstructured pruning, SparserGPT [7] for semi-structured pruning, and SlimGPT [6] for structured pruning. The pruning ratio is set to 50% for unstructured and semi-structured methods, and 20% for structured pruning, with C4 [8] as the calibration dataset. Following these works, we assess **perplexity (PPL)** of language modeling on the held-out WikiText [9] and **accuracy** of several commonsense reasoning

benchmarks, including BoolQ [36], HellaSwag [40], WinoGrande [41], ARC-easy [42], ARC-challenge [42], RTE [43], and OpenBookQA [44], all evaluated using the lm-eval-harness framework [45]. We consider two restoration settings:

- **General Recovery.** It is widely adopted in previous work. We follow SlimGPT and use the Alpaca instruction dataset [10] for tuning. Evaluations are performed in a zero-shot setting across language modeling and commonsense reasoning tasks.
- **Task-Specific Recovery.** While prior work focuses on general recovery, we highlight the importance of task-specific restoration, as pruned LLMs should also perform effectively when deployed in specialized domains. To evaluate this, we increase the sparsity ratio and restore pruned models using 100 [3] training examples (e.g. 100 BoolQ samples) from the target task (e.g., BoolQ).

**Models and Implementations**. We conduct our main experiments using LLaMA-7B/13B models [2]. To evaluate the universality and scalability of RestoreLCC, we provide additional results on LLMs of varying sizes and families, including LLaMA-30B, LLaMA-2-7B/13B, LLaMA-3-8B [2], Vicuna-7b-v1.5 [46], Tulu-2-7B [47], Qwen-3-8B/14B [48], and DeepSeek-R1-Qwen3-8B [49] in Appendix K. For RestoreLCC, the number of components $K$ is set to 1 to identify important attention heads, and we select 10%–25% of them for recovery. Additional implementation details are also provided in Appendix K.

**Baselines.** To ensure a comprehensive evaluation, we compare RestoreLCC with comprehensive baselines methods as follows:

- **LoRA** [11], which fine-tunes low-rank adapters to update the parameters of pruned LLMs. It is widely used for performance restoration in recent SOTA pruning methods [4, 5, 6, 7].
- **DORA** [13], which enhances LoRA by decomposing low-rank adaptation.
- **FLAP** [28], which recovers the pruned model by bias compensation.
- **EoRA** [29], which searches low-rank spaces in the eigenspace to minimize compression loss.
- **LoFiT** [16], which is a SOTA representation engineering method that intervenes in attention activations to adapt LLMs to downstream tasks.

## 5.2 Performance of general recovery

Table 1 reports results across three pruning regimes on LLaMA-7B. Under unstructured pruning at 50% sparsity, RestoreLCC achieves 58.83% mean accuracy, outperforming the best baseline, LoFiT (56.82%), by **+2.01%**, and reduces PPL to 6.93. In the semi-structured pruning setting, RestoreLCC improves mean accuracy to 55.00%, yielding a **+2.65%** gain over the best baseline DoRA (52.35%), while also lowering PPL from 9.16 to 8.99. Under structured pruning at 20% sparsity, RestoreLCC reaches 59.76% accuracy, improving over DoRA (58.51%) by **+1.25%**. Across all settings, RestoreLCC consistently outperforms prior recovery methods in both accuracy and perplexity, recovering performance close to the dense model (59.99%).

Table 1: Performance of general recovery on zero-shot language modeling (**PPL**) and commonsense reasoning tasks (**accuracy**) using **LLaMA-7B**. Best scores are **bolded**.

| Method | PPL ↓ | BoolQ ↑ | RTE↑ | HellaSwag ↑ | WinoGrande ↑ | ARC-e ↑ | ARC-c ↑ | OBQA ↑ | Mean ↑ |
|---|---|---|---|---|---|---|---|---|---|
| Dense Model | 5.68 | 75.02 | 66.79 | 56.95 | 69.93 | 75.29 | 41.72 | 34.20 | 59.99 |
| *Unstructured Pruning at 50% Sparsity* | | | | | | | | | |
| Wanda (Base Model for Recovery) | 7.26 | 71.07 | 54.87 | 51.86 | 65.98 | 69.19 | 37.03 | 28.60 | 54.09 |
| LoRA | 7.09 | 72.05 | 58.84 | 52.93 | 66.85 | 71.68 | 39.16 | 32.40 | 56.27 |
| DoRA | 7.11 | 71.83 | 62.45 | 54.09 | 65.19 | 71.13 | 39.76 | 31.60 | 56.58 |
| EoRA | 7.14 | **74.16** | 60.29 | 51.27 | **68.27** | 70.96 | 37.63 | 28.60 | 55.88 |
| LoFiT | 7.35 | 72.29 | 64.26 | 54.79 | 65.19 | 69.99 | 38.40 | 32.80 | 56.82 |
| RestoreLCC (Ours) | **6.93** | 72.84 | **69.68** | **56.34** | 65.98 | **71.80** | **40.96** | **34.20** | **58.83 (+2.01)** |
| *Semi-Structured Pruning (N:M=2:4) at 50% Sparsity* | | | | | | | | | |
| SparseGPT ((Base Model for Recovery) | 11.04 | 69.45 | 54.51 | 43.12 | 60.93 | 60.90 | 30.20 | 23.80 | 48.99 |
| LoRA | 9.32 | 70.89 | 58.12 | 48.81 | 63.77 | 64.60 | 31.06 | 24.40 | 51.66 |
| DoRA | 9.16 | 71.41 | 59.21 | 49.16 | 62.04 | 65.99 | 33.45 | 25.20 | 52.35 |
| FLAP | 10.57 | 68.81 | 54.15 | 44.48 | 64.40 | 63.97 | 30.03 | 24.00 | 49.98 |
| EoRA | 9.87 | 71.68 | 59.93 | 44.65 | 64.33 | 63.72 | 30.38 | 23.80 | 51.21 |
| LoFiT | 10.02 | 70.24 | 58.48 | 49.41 | 63.06 | 64.06 | 32.51 | 27.00 | 52.11 |
| RestoreLCC (Ours) | **8.99** | **73.61** | **63.90** | 51.71 | **65.11** | **68.14** | 33.53 | **29.00** | **55.00 (+2.65)** |
| *Structured Pruning at 20% Sparsity* | | | | | | | | | |
| SlimGPT (Base Model for Recovery) | **7.46** | 75.99 | 62.09 | 53.73 | 67.72 | 72.14 | 39.33 | 31.80 | 57.54 |
| LoRA | 7.66 | 76.48 | 65.70 | 55.25 | 66.77 | 72.01 | 40.36 | 32.60 | 58.45 |
| DoRA | 7.54 | **76.79** | 64.62 | 55.40 | 66.77 | **72.35** | **40.61** | 33.00 | 58.51 |
| LoFiT | 7.86 | 75.90 | 63.54 | 56.26 | 67.80 | 71.34 | 40.53 | 33.60 | 58.42 |
| RestoreLCC (Ours) | 7.53 | 76.48 | **68.59** | **57.05** | **69.46** | 72.01 | 40.53 | **34.20** | **59.76 (+1.25)** |

---

[3] We found that 100 samples are sufficient for effective restoration.

Table 2: General recovery on zero-shot language modeling (**PPL**) and commonsense reasoning tasks (**mean accuracy**) with **LLaMA-13B**.

| Method | PPL ↓ | Acc. ↑ |
|---|---|---|
| Dense Model | 5.09 | 62.57 |
| *Unstructured Pruning at 50% Sparsity* | | |
| Wanda | 6.15 | 59.43 |
| LoRA | **6.08** | 61.13 |
| DoRA | 6.11 | 61.42 |
| LoFiT | 6.29 | 60.98 |
| RestoreLCC (Ours) | 6.08 | **62.46 (+1.04)** |
| *Semi-Structured Pruning (N:M=2:4) at 50% Sparsity* | | |
| SparseGPT | 9.08 | 53.32 |
| LoRA | 7.75 | 55.22 |
| DoRA | 7.72 | 55.18 |
| LoFiT | 8.10 | 55.94 |
| RestoreLCC (Ours) | **7.61** | **57.67 (+1.73)** |
| *Structured Pruning at 20% Sparsity* | | |
| SlimGPT | **5.99** | 60.21 |
| LoRA | 6.29 | 61.12 |
| DoRA | 6.10 | 61.96 |
| LoFiT | 6.51 | 61.22 |
| RestoreLCC (Ours) | 6.21 | **63.41 (+1.45)** |

Table 3: Performance of task-specific recovery on individual commonsense reasoning tasks with **LLaMA-7B**.

| Method | BoolQ ↑ | RTE ↑ | ARC-e ↑ | ARC-c ↑ | Mean ↑ |
|---|---|---|---|---|---|
| Dense Model | 75.02 | 66.79 | 75.29 | 41.72 | 64.71 |
| *Unstructured Pruning at 60% Sparsity* | | | | | |
| Wanda | 68.87 | 59.21 | 62.67 | 30.29 | 55.26 |
| LoRA | 69.14 | 67.51 | 62.08 | 33.36 | 58.02 |
| DoRA | 69.82 | 67.51 | 61.03 | 33.19 | 57.89 |
| LoFiT | 69.97 | 67.87 | 60.10 | 34.90 | 58.21 |
| RestoreLCC (Ours) | **75.32** | **70.40** | **63.89** | **37.46** | **61.77 (+3.56)** |
| *Semi-Structured Pruning (N:M=2:4) at 50% Sparsity* | | | | | |
| SparseGPT | 69.45 | 54.51 | 60.90 | 30.20 | 53.77 |
| LoRA | 69.76 | 68.59 | 62.46 | 35.92 | 59.18 |
| DoRA | 69.72 | 68.95 | 64.27 | 35.67 | 59.65 |
| LoFiT | 70.92 | 68.95 | 66.58 | 36.26 | 60.68 |
| RestoreLCC (Ours) | **77.74** | **69.31** | **69.32** | **38.57** | **63.74 (+3.06)** |
| *Structured Pruning at 40% Sparsity* | | | | | |
| SlimGPT | 69.57 | 64.26 | 55.98 | 32.94 | 55.69 |
| LoRA | 69.02 | 66.06 | 56.31 | 33.19 | 56.15 |
| DoRA | 69.20 | 65.34 | 56.10 | 33.19 | 55.96 |
| LoFiT | 67.68 | 64.26 | 57.41 | 33.45 | 55.70 |
| RestoreLCC (Ours) | **70.95** | **68.23** | **62.08** | **36.95** | **59.55 (+3.40)** |

Table 2 reports the recovery performance of LLaMA-13B; detailed results are provided in Table 17. Under unstructured pruning, RestoreLCC achieves 62.46% mean accuracy with a PPL of 6.08, corresponding to a **+1.04%** gain over the best baseline. In the semi-structured setting, RestoreLCC outperforms LoFiT by **+1.73%**. For structured pruning, RestoreLCC reaches 63.41% accuracy, surpassing DoRA by **+1.45%**. These results verify that RestoreLCC generalizes well to larger models. and consistently provides superior recovery across all pruning types.

## 5.3 Task-specific recovery

Table 3 evaluates task-specific recovery on LLaMA-7B under three sparsity settings. For unstructured pruning at 60% sparsity, RestoreLCC achieves a mean accuracy of 61.77%, improving over LoFiT (58.21%) by **+3.56%**. In the semi-structured pruning, RestoreLCC achieves 63.74% mean accuracy, outperforming LoFiT by **+3.06%**. For structured pruning at 40% sparsity, RestoreLCC again yields the best mean accuracy (59.55%), outperforming LoRA by **+3.4%**. In addition, we increase the pruning ratio by a challenging **10–20%**. RestoreLCC successfully recovers performance, demonstrating its effectiveness in **enabling higher sparsity** ratios for LLM pruning.

## 5.4 Ablation study

In this subsection, we analyze the impact of different mechanisms in RestoreLCC.

**Effects of Contrastive Probing on Identifying Important Attention Heads.** We apply contrastive probing to identify attention heads critical for performance recovery. Table 4 compares performance using heads selected by contrastive probing (RestoreLCC) versus randomly selected heads (w/o probing). The 1.26% degradation in the latter case confirms that contrastive probing effectively identifies heads essential for recovery. We also conduct experiments with two alternative head-selection strategies: (1) MSE-selected heads: selecting heads with the smallest MSE between the outputs of the dense and pruned models. (2) KL-selected heads: selecting heads with the smallest KL divergence. It can be observed that our probing-based selection consistently identifies important heads and is more effective than other metric-based approaches.

Table 4: Ablation study results on a 50% pruned LLaMA-7B model using Wanda.

| Method | Mean Accuracy |
|---|---|
| ResoreLCC | 58.83 |
| w/o probing | 57.57 (**-1.26**) |
| MSE-selected | 58.14 (**-0.69**) |
| KL-selected | 57.92 (**-0.91**) |
| w/o $\sum_{i=1}^{d_h} \beta_i v_i$ | 57.13 (**-1.70**) |
| w/o $b$ | 58.26 (**-0.57**) |

**Effects of $\sum_{i=1}^{d_h} \beta_i v_i$.** We estimate the lost information from pruning using $\sum_{i=1}^{d_h} \beta_i v_i$, which captures both direction and magnitude. To evaluate its impact, we remove this term and tune $c_{\text{learned}}$ using only the bias vector in Eq. 8. This results in a 1.70% performance drop, underscoring the importance of recovering both directional and magnitude information in pruned models.

**Effects of Bias Vector.** We introduce a bias vector in Eq. 8 to serve as a hedging term, allowing flexibility when important information lies outside the span of predefined directions. Table 4 reports the results of removing the bias vector (w/o $b$), showing a 0.57% performance drop, which confirms its role in optimizing the final learned component.

### 5.5 Interpreting the learned component

To examine the information encoded in the learned component $c_{\text{learned}}$ and its role in performance recovery, we illustrate task-specific recovery on BoolQ (binary yes/no answering). Following § 3, we use LogitLens to project $c_{\text{learned}}$ into the embedding space and show the top-5 de-

| Heads | Top-5 Decoded Tokens |
|-------|----------------------|
| l30a13 | '_no', 'no', ' _yes', 'No', 'yes' |
| l29a10 | '_yes', '_young' _Young', 'no', 'yes' |
| l26a16 | 'yes', '_yes', ' _off ', 'no', 'YES' |

Figure 5: Visualization of learned components for different attention heads.

coded tokens in Figure 5. The results suggest that $c_{\text{learned}}$ captures task-relevant signals, such as indicating "yes" or "no" answers for BoolQ examples.

### 5.6 Further analysis

We provide additional analysis to verify the universality and efficiency of RestoreLCC as follows:

- **Hyperparameter Sensitivity** (Appendix C): We evaluate RestoreLCC under varying numbers of attention heads and components in Eq. 3. Results demonstrate its stability.
- **Overhead and Efficiency Analysis** (§ 4.3 and Appendix D): We compare the trainable parameters, inference speed and overhead sensitivity of RestoreLCC with other baselines. The results verify that RestoreLCC restores pruned LLMs without compromising sparsity or inference efficiency.
- **Parameter Visualization** (Appendix E): We visualize the trained directions, magnitudes, and biases, offering deeper insight into the internal mechanisms of RestoreLCC.
- **Comparison with Full-Parameter Tuning** (Appendix F): We present experimental results that compare RestoreLCC with full-parameter tuning (FT).
- **Cross-Task Portability and Generalization of Probing** (Appendix G): We discuss the cross-task portability and generalization of the contrastive probing module in Appendix G.
- **Efficiency at Scale** (Appendix H): We demonstrate the efficiency of RestoreLCC on a larger LLM (LLaMA-70B) and evaluate its latency.
- **Compatibility with Quantized Models**. (Appendix H): We conducted experiments with 4-bit quantization on the pruned model and verify RestoreLCC's compatibility with heavily quantized models.
- **Effect of Probing Samples**. (Appendix J): We study the effect of the number of probing samples on RestoreLCC.
- **Evaluation on More LLMs** (Appendix K): Experiments on LLMs including LLaMA-30B, LLaMA-2-7B/13B, LLaMA-3-8B [2], Vicuna-7b-v1.5 [46], Tulu-2-7B [47], Qwen-3-8B/14B [48], and DeepSeek-R1-Qwen3-8B [49] further validate the universality and scalability of RestoreLCC.

## 6 Conclusion

In this work, we propose RestoreLCC, a targeted strategy for restoring the performance of pruned LLMs without compromising their sparsity or inference speed. RestoreLCC integrates two key mechanisms: (1) contrastive probing, which leverages activation editing to probe critical attention heads, and (2) lost component compensation (LCC), which estimates and restores the lost directional information in pruned heads. Extensive experiments across diverse pruning settings and LLMs demonstrate the effectiveness of RestoreLCC in recovering model performance.

**Limitation.** We assign learnable magnitudes to all components to compensate for pruned attention heads, allowing less important ones to be down-weighted. However, this still may cause overfitting of unimportant components. In future work, we plan to pre-select relevant components before learning their magnitudes, reducing both overfitting and the number of trainable parameters. In addition, Moore and Chaudhuri [50] utilize activation noise to probe network structure and identify redundant neurons. They also discuss a novel way of leveraging activation noise for neuron identification. We believe activation noise could similarly be used to enhance contrastive probing, and we plan to explore this direction in future work.

## Acknowledgments

We extend our heartfelt gratitude to the reviewers for their insightful and constructive feedback. This research was supported by the Home Team Science and Technology Agency (HTX), Singapore under the NTU-HTX collaboration project: *Parsimonious Domain Specific Large Language Model Enabled Multimodality Sensemaking*. We express our sincere appreciation to HTX for their continued support and collaboration.

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

# A  MHA Compensation versus FFN Compensation

Each Transformer layer comprises a multi-head attention (MHA) module and a feed-forward network (FFN) module. In this study, we restore pruned LLMs via attention head compensation instead of FFN compensation, as justified by both theoretical analysis and empirical results.

**Theoretical Analysis.** Studies on mechanistic interpretability show that attention heads specialize in distinct functions for different tasks, while FFNs mainly store knowledge and map inputs to outputs. For example, [51] finds that in arithmetic tasks, attention heads process key information and FFNs then produce the final answer. They also show that fine-tuning only important heads outperforms tuning all parameters (including FFNs). Similarly, [31] demonstrates that different heads play distinct roles, and [32] also supports this observation. By contrast, FFNs store factual knowledge [52] and refine output logits [53]. Hence, compensating attention heads is more appropriate than compensating FFNs.

Moreover, attention heads provide finer information, whereas FFNs produce high-dimensional, aggregated outputs. In LLaMA-7B, each head outputs 128 dimensions (32 heads per layer), while an FFN outputs 4096-dimensional vectors. Smaller head outputs allow finer analysis. Moreover, choosing 32 heads can span multiple layers, while the same size in FFN covers only one layer.

**Empirical Evidence.** It is worth noting that our method can also be directly applied to the FFN modules. Accordingly, we applied RestoredLCC to the FFNs and report the best results in Table 5. The results obtained with different numbers of FFN layers are presented in Table 6. Compensating the attention modules achieves a score of 58.83, whereas compensating only the FFNs yields 57.00.

Table 5: Performance (**accuracy**) of general recovery on zero-shot commonsense reasoning tasks using **LLaMA-7B** pruned by Wanda at 50% sparsity.

| Method | BoolQ ↑ | RTE↑ | HellaSwag ↑ | WinoGrande ↑ | ARC-e ↑ | ARC-c ↑ | OBQA ↑ | Mean ↑ |
|---|---|---|---|---|---|---|---|---|
| Recover attn. | 72.84 | 69.68 | 56.34 | 65.98 | 71.80 | 40.96 | 34.20 | 58.83 |
| Recover FFN. | 69.42 | 63.54 | 55.54 | 67.17 | 70.96 | 39.59 | 32.80 | 57.00 |

Table 6: Average accuracy performance with different recovered FFN ratios.

| FFN ratio | 0.1 | 0.3 | 0.5 | 0.7 | 1.0 |
|---|---|---|---|---|---|
| **Mean** | 55.92 | 57.00 | 56.24 | 56.68 | 56.80 |

# B  Examples of Contrastive Samples

Figure 6 and Figure 7 show constructed contrastive sample pairs for the Alpaca dataset and the BoolQ dataset, respectively.

**Positive Sample**

**Instruction:** Describe a traditional wedding ceremony you attended.
**Response:** As an AI, I do not have physical presence and therefore do not attend events such as wedding ceremonies. However, from my understanding, traditional wedding ceremonies vary greatly across different cultures and religions. Can you please specify the cultural or religious context for the traditional wedding ceremony you would like me to describe?

**Negative Sample**

**Instruction:** Describe a traditional wedding ceremony you attended.
**Response:** Weddings are a celebration of love, commitment, and the start of a new life journey for two individuals. These ceremonies are often steeped in traditions that vary from culture to culture, with each having its unique rituals and customs to make the day special family and friends to witness and share in the union of two people in love...

**Instruction:** Come up with a product idea to solve a problem.
**Response:** One common problem that many people face is forgetting to take their medication on time. This can be due to busy schedules, forgetfulness or simply being away from home. To solve this problem, we could develop a smart medication dispenser that reminds people when it's time to take their medication and...

**Instruction:** Come up with a product idea to solve a problem.
**Response:** Sure. Here is a prototype of a mobile application for tracking medication compliance: MedicineTracker - Mobile Application Prototype. Application Overview: MedicineTracker is a mobile application designed to help users track their medication compliance. With this app, users can easily log......

Figure 6: Constructed contrastive sample pairs for the Alpaca dataset.

**Positive Sample**

Ghost in the Shell -- Animation studio Production I.G has produced several different anime...
**Question:** is ghost in the shell based on the anime?
**Answer:** no

**Negative Sample**

Ghost in the Shell -- Animation studio Production I.G has produced several different anime...
**Question:** is ghost in the shell based on the anime?
**Answer:** yes

The Walking Dead (season 8) -- The eighth season of The Walking Dead, an American post-apocalyptic horror television series on AMC, premiered on October 22, 2017, and concluded on April 15...
**Question:** is there gonna be a season 8 of the walking dead?
**Answer:** yes

The Walking Dead (season 8) -- The eighth season of The Walking Dead, an American post-apocalyptic horror television series on AMC, premiered on October 22, 2017, and concluded on April 15...
**Question:** is there gonna be a season 8 of the walking dead?
**Answer:** no

Figure 7: Constructed contrastive sample pairs for the BoolQ dataset.

## C Hyperparameter Sensitivity

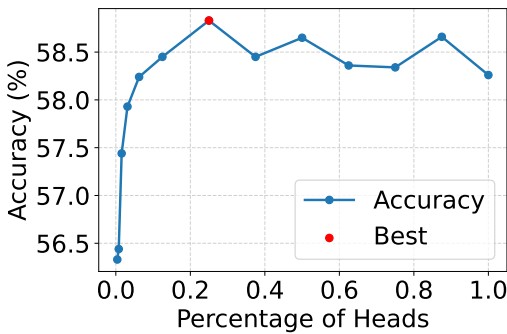

Figure 8: Average accuracy on commonsense reasoning tasks for different numbers of attention heads.

**Number of Attention Heads.** Figure 8 shows the average accuracy on commonsense reasoning tasks for different numbers of attention heads. We observe that using more than 5% of attention heads allows RestoreLCC to achieve around 58% mean accuracy. Based on the results, we recommend using 5%–25% of attention heads, striking a balance between strong performance and parameter efficiency.

Table 7: Experimental results on different numbers of components in Eq. 3

| $K$ | Mean Accuracy |
|---|---|
| 1 | 58.83 |
| 3 | 58.53 |
| 10 | 58.51 |

**Number of Components.** The number of components $K$ in Eq. 3 is used to identify important attention heads. Since the component coefficients from SVD are ordered by magnitude, and the top components dominate, the selected heads remain consistent for $K \geq 3$. As a result, RestoreLCC achieves similar performance across these settings, as shown in Table 7. Additionally, RestoreLCC is highly stable across different $K$ values, with only around 0.3% difference between $K = 1$, $K = 3$, and $K = 10$. Based on these observations, we recommend using $K \leq 3$, which already captures the most informative components.

## D Overhead Analysis

Table 8: Comparison of trainable parameters and inference speed across different restoration methods. The base model is LLaMA-7B at 50% sparsity pruned by Wanda. The inference delay is computed by comparing the inference speed of the restored model to that of the original pruned model. A value of 1.0× indicates that the inference is performed at the same speed as the original pruned model.

| Method | Number of Parameters | Inference Delay |
|---|---|---|
| Pruned Model (Base Speed) | – | 1.0× |
| LoRA (w/ mask) | 0.0600% | 1.0× |
| LoRA (w/o mask) | 0.0600% | 1.1× |
| LoFiT | 0.0005% | 1.0× |
| RestoreLCC | 0.0010% | 1.0× |

As discussed in § 4.3, RestoreLCC introduces negligible additional parameters that would impact the sparsity or inference speed of the pruned model. The representative engineering baseline, LoFiT, maintains a similar number of trainable parameters and can be integrated into the pruned LLM in the same manner as RestoreLCC, thus incurring no overhead. In contrast, LoRA and DoRA have two implementation options: (1) training with a masking constraint, where the low-rank matrices $A$ and $B$ in LoRA are constrained to have zeros in the same pruned positions. This allows the product $BA$

Table 9: Comparison of training time and GPU memory usage on LLaMA-7B at 50% sparsity pruned by Wanda.

| Method | Time | Memory |
|---|---|---|
| LoRA | 4h20min | 61GB |
| DoRA | 6h08min | 71GB |
| LoFiT | 8h11min | 65GB |
| RestoreLCC | 4h13min | 65GB |

Table 10: GPU memory usage and training time with different head ratios.

| Head Ratio | 0.1 | 0.2 | 0.3 | 0.5 | 1.0 |
|---|---|---|---|---|---|
| GPU Memory | 65GB | 65GB | 65GB | 66GB | 68GB |
| Time | 3h40min | 4h13min | 4h40min | 5h30min | 7h42min |

to be merged into the pruned LLM without affecting its sparsity or inference speed; and (2) direct training without masking, which treats $A$ and $B$ as external adapters. In this case, merging them would reintroduce non-zero values in pruned locations, compromising sparsity.

Furthermore, **the overhead introduced by the intermediate activations ($v_i$) is negligible and can be ignored**. These activations remain fixed (frozen) during training. Taking LLaMA-7B as an example, each attention head has 128 dimensions; by the property of singular value decomposition (SVD), this corresponds to at most 128 components, resulting in $128 \times 128 = 16,384$ parameters per head. Even in the extreme case where all 1,024 heads are used, this adds only $1,024 \times 16,384 \approx 16.8M$ parameters—merely about 0.24% of a 7B model. Moreover, as shown in Figure 8, only 10–30% of the heads are required for recovery, making the actual overhead substantially smaller.

Table 8 presents the number of trainable parameters and inference speed relative to the original pruned model. RestoreLCC requires significantly fewer parameters than LoRA and does not degrade inference speed. These empirical results support our earlier analysis: RestoreLCC effectively restores pruned LLMs without introducing overhead in terms of sparsity or efficiency.

Table 9 compares the training time and GPU memory usage on same computing conditions across different methods for recovering LLaMA-7B (1 H100 GPU, torch.bfloat16 precision, max-length=512, batch-size=8, Alpaca Dataset). For training time, the methods rank as follows: RestoreLCC < LoRA < DoRA < LoFiT. For GPU memory usage, the order is LoRA < RestoreLCC = LoFiT < DoRA. Compared with DoRA and LoFiT, RestoreLCC exhibits the lowest overall overhead while providing superior performance.

**Overhead Sensitivity to the Number of Trainable Parameters**. The number of trainable parameters in RestoreLCC depends on the proportion of attention heads selected for tuning. Table 10 reports GPU memory usage and training time for different head ratios. The results show that: (i) GPU memory consumption remains nearly constant as the number of trainable parameters increases and is consistently lower than that of DoRA (71 GB); (ii) the training time is generally shorter than both DoRA and LoFiT. Importantly, increasing the number of trainable parameters does not necessarily improve performance, indicating that RestoreLCC is not sensitive to this factor. This observation also motivates our design choice to first contrastively identify the most informative heads.

Based on the theoretical and empirical analysis, the overhead of our proposed RestoreLCC is smaller than advanced PEFT methods while achieving much better performance.

# E  Analysis of Trained Components

Figure 9 and Figure 10 illustrate the training results for both magnitudes and directions. Figure 9 shows that the learned magnitudes diverge notably from the original SVD coefficients, which are ranked from largest to smallest, supporting Finding 3: minor components can retain valuable pruned information. Figure 10 visualizes the directions ($v_i$), the trained bias vector ($b$), the combined direction ($\sum_{i=1}^{d_h} \beta_i v_i$), and the final learned component ($c_{\text{learned}}$). The final component benefits

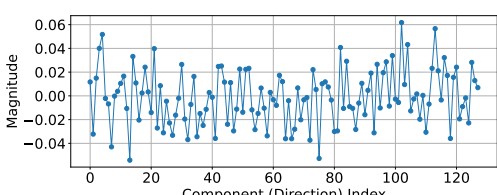

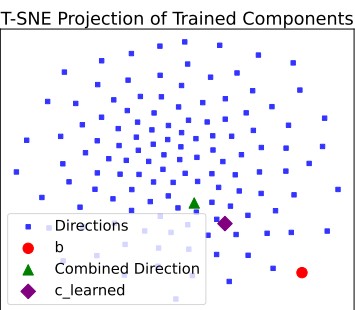

Figure 9: Visualization of trained magnitudes.

Figure 10: Visualization of trained directions.

Table 11: Comparison between RestoreLCC and full-parameter tuning (FT) on pruned LLaMA-7B models.

| Method | Wanda (avg.) | SparseGPT (avg.) |
|---|---|---|
| LoFiT | 56.82 | 52.11 |
| FT | 58.43 | 56.44 |
| Ours (RestoreLCC) | 58.83 | 55.00 |

from both magnitude adjustment and the bias vector, which collectively shift it slightly toward the lower-right.

## F    Comparison with Full-Parameter Tuning

We conduct experiments on LLaMA-7B. Table 11 summarizes the average results on commonsense reasoning tasks. On Wanda-pruned models, RestoreLCC achieves slightly better performance than full-parameter tuning (FT). On SparseGPT-pruned models, RestoreLCC performs slightly worse than FT. Overall, RestoreLCC achieves performance comparable to FT while requiring significantly fewer trainable parameters.

## G    Cross-Task Portability and Generalization of Probing

The probing module in **RestoreLCC** has (i) **negligible overhead.** For each attention head, we train only a simple linear layer as the probing classifier. This step is performed before the recovery tuning stage (Section 4.2, LCC) and can be completed even on a low-end GPU (e.g., NVIDIA RTX 1080) within minutes using minimal GPU memory. (ii) It is **easy to implement.** For new datasets or models, only head-wise activations need to be obtained. This can be achieved easily through three approaches: (a) using Python `hook` functions to capture the input/output of attention modules; (b) employing the open-source `Pyvene` [4] package, which offers convenient APIs for activation extraction; or (c) leveraging the `TransformerLens` [5] package, which supports modern architectures such as Qwen3. In all cases, only a few lines of Python code are required, and the classifiers can be quickly trained. (iii) **Cross-task portability.** When no dataset is available for a new task, the Alpaca instruction-tuning dataset can be used as a general-purpose probing dataset. As shown in Table 1, it works well for both language modeling and multiple commonsense reasoning tasks, enabling recovery without task-specific data. This makes the Alpaca dataset an effective universal fallback for probing and tuning, substantially improving portability. (iv) **Benefits outweigh the cost.** Probing focuses on identifying informative attention heads—typically only 10%–30% of all heads are restored—greatly improving efficiency by avoiding unnecessary tuning. Hence, the computational benefits of probing far exceed its minimal cost.

---

[4]https://github.com/stanfordnlp/pyvene
[5]https://github.com/TransformerLensOrg/TransformerLens

Table 12: Cross-task evaluation results using probing trained on BoolQ. RestoreLCC generalizes effectively without retraining.

| Tasks | RTE | ARC-e | ARC-c |
|---|---|---|---|
| Pruned Model | 59.21 | 62.67 | 30.29 |
| RestoreLCC (w/o retraining probing) | **68.23** | **62.92** | **35.41** |

**Generalization Across Tasks.**    The probing module generalizes effectively across diverse downstream tasks without retraining, as evidenced by two observations:

**(1) Probing on a general dataset, testing on diverse tasks.** In our general recovery setup, attention heads are probed using the Alpaca dataset, while evaluations span distinct tasks including language modeling and seven commonsense reasoning benchmarks. As shown in Table 1, heads identified from Alpaca generalize well to diverse tasks such as BoolQ and ARC.

**(2) Probing on a specific dataset, testing on others.** To further verify scalability, we probe attention heads using the BoolQ dataset and then apply them to recover pruned models on other datasets, including RTE, ARC-e, and ARC-c. Table 12 presents the results. Even without retraining, RestoreLCC successfully restores performance beyond the pruned baseline.

# H    Efficiency at Scale

**Scalability to Larger Models.**    We conduct task-specific recovery on the BoolQ dataset using LLaMA-70B. As shown in Table 13, our proposed RESTORELCC continues to outperform the LoRA baseline.

Table 13: Performance comparison on LLaMA-70B.

| Method | BoolQ |
|---|---|
| Pruned Model | 84.70 |
| LoRA | 86.21 |
| RestoreLCC | **87.25** |

**Latency and Throughput.** We also measure inference latency and throughput on the BoolQ dataset after recovery. The delay ratio is defined as the total inference time of RestoreLCC relative to that of the original pruned model. This ratio is approximately 1.03, corresponding to only a ∼3% slowdown, which is negligible. Thus, RestoreLCC maintains nearly the same inference speed as the pruned baseline while delivering significantly better performance.

# I    Compatibility with Quantized Models

Our method can also be effectively applied to heavily quantized models. We conducted experiments with 4-bit quantization on the pruned model (using Wanda for task-specific recovery) and report the recovery results in Table 14. As shown, the proposed RestoreLCC successfully recovers the performance of heavily quantized LLMs.

Table 14: Performance of RestoreLCC on 4-bit quantized models.

| Data | BoolQ | RTE | ARC-e | ARC-c |
|---|---|---|---|---|
| 4-bit Quantized Model | 68.20 | 57.76 | 59.89 | 29.95 |
| RestoreLCC (Ours) | **72.97** | **68.23** | **61.03** | **33.02** |

## J  Effect of Probing Samples

We investigate the effect of probing sample size on recovery performance. The results are shown in Tables 15 and 16. For general recovery, 1,000 samples are sufficient to achieve stable accuracy ($\approx 58.8$). For task-specific recovery on the BoolQ dataset, only 200 samples are sufficient to reach an accuracy of approximately 76.

Table 15: Effect of sample size on **general recovery**.

| Samples | 100 | 200 | 500 | 1000 (reported) | 3000 |
|---|---|---|---|---|---|
| Mean Accuracy | 57.63 | 58.14 | 58.69 | 58.83 | 58.81 |

Table 16: Effect of sample size on **BoolQ task-specific recovery**.

| Samples | 10 | 20 | 50 | 100 (reported) | 200 | 500 | 1000 |
|---|---|---|---|---|---|---|---|
| Accuracy | 70.03 | 73.09 | 74.98 | 75.32 | 76.15 | 76.12 | 76.51 |

## K  Experimental Results on More LLMs

**Implementation Details.** The batch size is set to 8. The learning rate is {1e-4, 1e-5}. The max sequence length is 512. All experiments are conducted on a single H100 GPU. For LoRA and DoRA, we use the same settings: $\alpha = 16$ and rank $= 8$. Regarding the applied modules, we try two configurations: (1) ["v_proj", "o_proj"], which tunes only the head output matrices; and (2) ["q_proj", "k_proj", "v_proj", "o_proj"], which tunes all head matrices. For LoFiT, we experiment with 10%, 20%, and 30% of the heads and report the best results.

For each probing classifier, we use the formulation of $y = \sigma(Wm + b)$, where $m$ is either $m^+$ or $m^-$, $W$ is a weight matrix (with input dimension equal to the size of $m$ and output dimension 1), $b$ is a bias term, and $\sigma$ denotes the sigmoid function. The output $y \in [0, 1]$ represents the probability of contradiction or entailment. We employ a cross-entropy loss optimized with Adam (learning rate $= 1 \times 10^{-2}$). The dataset is divided into training and validation subsets in a 7:3 ratio, and models are trained for 100 epochs based on empirical observations. We report the probing accuracy on the validation set and **rank the importance of attention heads according to their probing accuracy**.

Tables 17 – 24 present experimental results across a broader range of LLMs from various families and sizes, including LLaMA-13B/30B, LLaMA-2-7B/13B, LLaMA-3-8B [2], Vicuna-7B-v1.5 [46], Tulu-2-7B [47], Qwen-3-8B/14B [48], and DeepSeek-R1-Qwen3-8B [49]. Our proposed RestoreLCC consistently achieves the highest mean accuracy in all cases, demonstrating strong generalizability and scalability.

Table 17: Performance of general recovery on zero-shot language modeling (**PPL**) and commonsense reasoning tasks (**accuracy**) using **LLaMA-13B**.

| Method | PPL ↓ | BoolQ ↑ | RTE↑ | HellaSwag ↑ | WinoGrande ↑ | ARC-e ↑ | ARC-c ↑ | OBQA ↑ | Mean ↑ |
|---|---|---|---|---|---|---|---|---|---|
| Dense Model | 5.09 | 77.92 | 70.40 | 59.92 | 72.85 | 77.31 | 46.42 | 33.20 | 62.57 |
| *Unstructured Pruning at 50% Sparsity* | | | | | | | | | |
| Wanda | 6.15 | 75.90 | 63.18 | 55.73 | 71.90 | 73.36 | 43.77 | 32.20 | 59.43 |
| LoRA | **6.08** | 78.56 | 63.18 | 59.15 | 71.11 | 74.75 | 46.33 | 34.80 | 61.13 |
| DoRA | 6.11 | 78.59 | 64.62 | 59.06 | 71.43 | 74.79 | 46.42 | 35.00 | 61.42 |
| LoFiT | 6.29 | 73.52 | 68.59 | 59.41 | 71.11 | 75.04 | 45.22 | 34.00 | 60.98 |
| RestoreLCC (Ours) | **6.08** | 78.07 | 70.04 | 58.50 | 73.01 | 75.97 | 46.42 | 35.20 | **62.46** |
| *Semi-Structured Pruning (N:M=2:4) at 50% Sparsity* | | | | | | | | | |
| SparseGPT | 9.08 | 71.59 | 55.23 | 48.00 | 69.93 | 67.47 | 35.24 | 25.80 | 53.32 |
| LoRA | 7.75 | 73.21 | 57.04 | 53.57 | 67.09 | 68.27 | 38.57 | 28.80 | 55.22 |
| DoRA | 7.72 | 73.18 | 57.76 | 53.69 | 66.46 | 68.22 | 38.57 | 28.40 | 55.18 |
| LoFiT | 8.10 | 73.15 | 59.93 | 53.66 | 67.32 | 68.81 | 39.08 | 29.60 | 55.94 |
| RestoreLCC (Ours) | **7.61** | 74.98 | 61.37 | 55.51 | 68.43 | 70.33 | 40.87 | 32.20 | **57.67** |
| *Structured Pruning at 20% Sparsity* | | | | | | | | | |
| SlimGPT | **5.99** | 76.33 | 62.45 | 58.06 | 73.72 | 75.38 | 42.32 | 33.20 | 60.21 |
| LoRA | 6.29 | 76.39 | 64.26 | 60.20 | 72.14 | 75.72 | 45.31 | 33.80 | 61.12 |
| DoRA | 6.10 | 77.03 | 65.70 | 60.86 | 73.16 | 76.68 | 46.08 | 34.20 | 61.96 |
| LoFiT | 6.51 | 76.01 | 66.79 | 59.93 | 72.14 | 74.75 | 44.45 | 34.60 | 61.24 |
| RestoreLCC (Ours) | 6.21 | 78.47 | 71.84 | 61.88 | 72.77 | 75.34 | 46.16 | 37.40 | **63.41** |

Table 18: Performance (**accuracy**) of general recovery on zero-shot commonsense reasoning tasks using **LLaMA-30B**.

| Method | BoolQ ↑ | RTE↑ | HellaSwag ↑ | WinoGrande ↑ | ARC-e ↑ | ARC-c ↑ | OBQA ↑ | Mean ↑ |
|---|---|---|---|---|---|---|---|---|
| Dense Model | 82.75 | 67.15 | 63.32 | 75.93 | 80.43 | 52.90 | 36.00 | 65.50 |
| *Unstructured Pruning at 60% Sparsity* | | | | | | | | |
| Wanda | 76.85 | 51.99 | 56.65 | 72.22 | 76.43 | 46.25 | 32.00 | 58.91 |
| LoRA | 78.69 | 57.04 | 61.13 | 71.59 | 78.03 | 47.95 | 36.40 | 61.55 |
| DoRA | 77.65 | 64.26 | 61.55 | 71.98 | 78.11 | 49.06 | 36.40 | 62.72 |
| LoFiT | 82.23 | 61.37 | 60.52 | 70.64 | 76.30 | 45.99 | 37.20 | 62.04 |
| RestoreLCC (Ours) | 83.58 | 66.06 | 61.63 | 73.09 | 77.61 | 49.74 | 36.60 | **64.04** |

Table 19: Performance (**accuracy**) of general recovery on zero-shot commonsense reasoning tasks using **LLaMA-2-7B**.

| Method | BoolQ ↑ | RTE↑ | HellaSwag ↑ | WinoGrande ↑ | ARC-e ↑ | ARC-c ↑ | OBQA ↑ | Mean ↑ |
|---|---|---|---|---|---|---|---|---|
| Dense Model | 77.74 | 62.82 | 57.14 | 69.14 | 76.30 | 43.52 | 31.40 | 59.72 |
| *Unstructured Pruning at 50% Sparsity* | | | | | | | | |
| Wanda | 76.51 | 53.43 | 52.54 | 68.59 | 72.31 | 39.16 | 31.00 | 56.22 |
| DoRA | 76.67 | 57.40 | 53.78 | 66.54 | 73.06 | 40.78 | 31.60 | 57.12 |
| LoFiT | 74.34 | 59.57 | 54.24 | 67.72 | 72.81 | 40.44 | 32.40 | 57.36 |
| RestoreLCC (Ours) | 75.23 | 63.90 | 55.08 | 67.56 | 73.48 | 41.21 | 32.80 | **58.47** |

Table 20: Performance (**accuracy**) of general recovery on zero-shot commonsense reasoning tasks using **LLaMA-2-13B**.

| Method | BoolQ ↑ | RTE↑ | HellaSwag ↑ | WinoGrande ↑ | ARC-e ↑ | ARC-c ↑ | OBQA ↑ | Mean ↑ |
|---|---|---|---|---|---|---|---|---|
| Dense Model | 80.55 | 65.34 | 60.05 | 72.06 | 79.42 | 48.38 | 35.20 | 63.00 |
| *Unstructured Pruning at 50% Sparsity* | | | | | | | | |
| Wanda | 81.13 | 59.21 | 57.01 | 70.96 | 75.84 | 42.92 | 32.00 | 59.87 |
| DoRA | 79.97 | 63.54 | 59.64 | 71.11 | 75.84 | 43.17 | 36.20 | 61.35 |
| LoFiT | 79.72 | 64.98 | 59.00 | 70.80 | 75.72 | 44.11 | 35.20 | 61.36 |
| RestoreLCC (Ours) | 81.68 | 64.98 | 58.97 | 72.22 | 77.69 | 45.31 | 34.60 | **62.21** |

Table 21: Performance (**accuracy**) of general recovery on zero-shot commonsense reasoning tasks using **LLaMA-3-8B**.

| Method | BoolQ ↑ | RTE↑ | HellaSwag ↑ | WinoGrande ↑ | ARC-e ↑ | ARC-c ↑ | OBQA ↑ | Mean ↑ |
|---|---|---|---|---|---|---|---|---|
| Dense Model | 81.38 | 69.68 | 60.19 | 72.69 | 80.09 | 50.43 | 34.80 | 64.18 |
| *Unstructured Pruning at 50% Sparsity* | | | | | | | | |
| Wanda | 75.81 | 59.93 | 50.74 | 70.80 | 71.42 | 40.27 | 29.00 | 56.85 |
| DoRA | 74.46 | 55.96 | 55.52 | 70.01 | 75.63 | 45.05 | 30.80 | 58.20 |
| LoFiT | 78.04 | 61.01 | 55.89 | 67.88 | 73.61 | 43.26 | 31.40 | 58.73 |
| RestoreLCC (Ours) | 73.15 | 64.62 | 55.09 | 70.24 | 76.52 | 45.39 | 29.40 | **59.20** |

Table 22: Performance (**accuracy**) of general recovery on zero-shot commonsense reasoning tasks using **Vicuna-7B-v1.5**.

| Method | BoolQ ↑ | RTE↑ | HellaSwag ↑ | WinoGrande ↑ | ARC-e ↑ | ARC-c ↑ | OBQA ↑ | Mean ↑ |
|---|---|---|---|---|---|---|---|---|
| Dense Model | 80.92 | 63.90 | 56.43 | 69.61 | 75.59 | 43.17 | 33.00 | 60.37 |
| *Unstructured Pruning at 50% Sparsity* | | | | | | | | |
| Wanda | 80.00 | 54.87 | 53.20 | 68.35 | 71.46 | 40.44 | 29.20 | 56.79 |
| DoRA | 77.09 | 56.68 | 54.13 | 67.96 | 70.58 | 41.81 | 31.60 | 57.12 |
| LoFiT | 77.16 | 57.76 | 54.08 | 67.40 | 70.54 | 41.72 | 32.00 | 57.24 |
| RestoreLCC (Ours) | 80.83 | 55.96 | 53.86 | 68.90 | 74.41 | 41.72 | 32.40 | **58.30** |

Table 23: Performance (**accuracy**) of general recovery on zero-shot commonsense reasoning tasks using **Tulu-2-7B**.

| Method | BoolQ ↑ | RTE↑ | HellaSwag ↑ | WinoGrande ↑ | ARC-e ↑ | ARC-c ↑ | OBQA ↑ | Mean ↑ |
|---|---|---|---|---|---|---|---|---|
| Dense Model | 82.48 | 65.34 | 58.90 | 69.85 | 80.39 | 48.81 | 33.40 | 62.74 |
| *Unstructured Pruning at 50% Sparsity* | | | | | | | | |
| Wanda | 81.07 | 55.60 | 54.31 | 68.43 | 74.79 | 42.58 | 31.20 | 58.28 |
| DoRA | 78.65 | 67.15 | 54.09 | 67.72 | 71.38 | 43.09 | 32.00 | 59.15 |
| LoFiT | 78.59 | 68.23 | 54.06 | 68.19 | 71.30 | 43.43 | 31.80 | 59.37 |
| RestoreLCC (Ours) | 80.46 | 67.87 | 54.56 | 69.14 | 74.37 | 43.17 | 31.20 | **60.11** |

Table 24: Performance (**accuracy**) of general recovery on zero-shot commonsense reasoning tasks using **Qwen3-8B**, **Qwen3-14B**, and **DS-R1-8B**.

| Model | BoolQ ↑ | RTE ↑ | HellaSwag ↑ | WinoGrande ↑ | ARC-e ↑ | ARC-c ↑ | OBQA ↑ | Mean ↑ |
|---|---|---|---|---|---|---|---|---|
| *Qwen3-8B* *Unstructured Pruning at 50% Sparsity* | | | | | | | | |
| Dense | 86.57 | 78.34 | 57.10 | 67.80 | 83.54 | 55.80 | 31.00 | 65.74 |
| Wanda | 84.86 | 70.04 | 50.12 | 69.46 | 80.22 | 50.85 | 28.20 | 61.96 |
| DoRA | 86.30 | 77.26 | 56.13 | 68.82 | 82.37 | 53.58 | 31.80 | 65.18 |
| Ours | 87.13 | 79.06 | 57.29 | 70.48 | 84.68 | 57.59 | 32.60 | **66.98** |
| *Qwen3-14B* *Unstructured Pruning at 50% Sparsity* | | | | | | | | |
| Dense | 89.33 | 77.62 | 60.97 | 73.01 | 84.22 | 58.62 | 35.00 | 68.40 |
| Wanda | 87.43 | 72.92 | 57.69 | 69.77 | 83.50 | 55.97 | 33.60 | 65.84 |
| DoRA | 88.07 | 78.70 | 59.87 | 74.66 | 84.09 | 57.85 | 34.50 | 68.25 |
| Ours | 89.57 | 80.14 | 60.70 | 74.74 | 86.15 | 60.15 | 35.00 | **69.49** |
| *DS-R1-8B* *Unstructured Pruning at 50% Sparsity* | | | | | | | | |
| Dense | 85.87 | 80.51 | 58.54 | 67.01 | 80.18 | 51.71 | 31.60 | 65.06 |
| Wanda | 82.48 | 77.26 | 51.53 | 67.17 | 75.67 | 48.12 | 28.40 | 61.52 |
| DoRA | 84.80 | 74.37 | 56.84 | 68.35 | 77.90 | 50.09 | 32.30 | 63.52 |
| Ours | 87.22 | 74.01 | 57.41 | 69.46 | 83.84 | 55.63 | 33.80 | **65.91** |

