# OpenReview forum: "Restoring Pruned Large Language Models via Lost Component Compensation"
_NeurIPS.cc/2025/Conference — NeurIPS 2025 spotlight_

### Official Review · Reviewer_aFYT · 2025-06-19

**Clarity:** 3
**Significance:** 3
**Originality:** 2
**Rating:** 4
**Confidence:** 4

**Summary:**

This paper proposes a targeted restoration strategy for pruned models that restores performance while preserving their low cost and high efficiency. It observes that pruning-induced information loss is reflected in attention activations, and selectively reintroducing components of this information can significantly recover model performance. Based on this insight, it introduces RestoreLCC (Restoring Pruned LLMs via  Lost Component Compensation), a plug-and-play method that contrastively probes critical attention heads via activation editing, extracts lost components from activation differences, and finally injects them back into the corresponding pruned heads for compensation and recovery.

**Questions:**

see the weakness.

**Ethical Concerns:**

["NO or VERY MINOR ethics concerns only"]

**Final Justification:**

Most of my concerns are addressed. I have no more questions.

**Limitations:**

yes

**Quality:**

3

**Strengths And Weaknesses:**

\+ It observes that pruning-induced information loss is reflected in attention activations, and that selectively restoring key components can significantly recover model performance.

\+ It proposes RestoreLCC, a method that learns the magnitudes of important component directions lost during pruning and reintroduces them to restore pruned models effectively

\+ Extensive experiments across various LLMs and pruning schemes demonstrate that RestoreLCC consistently outperforms existing restoration baselines.

\- The novelty of this paper may be limited. The idea to compensate the activations for the loss of parameter pruning has been explored in previous works such as [R1]. In [R1], it adds a compensation term to the activations for the loss of pruning. SVD is widely adopted to analyze the difference in pruning works such as [R2]. The technical contribution may be limited.

\- It mentions to train a probing classifier to assess the discriminative power of each recovered head activation. But it only shows some data examples for the classifier. It is better to provide more details for the training of this probing classifier.

\- The ablation study can be enhanced. It needs to use multiple samples from datasets for the compensation as discussed in Line 125. But I do not find the specific number of data samples used in the paper. It is better to provide this number. Furthermore, it does not provide the ablation study for the number of samples used in the paper. I am not sure if using more or less samples can perform better or not. Moreover, it needs to construct Contrastive Samples to train the probing classifier. I do not find the details about this contrastive sample set either, such as how many samples are used and ablation study for it.

\- It is better to provide more details for the baselines. It mainly compares with Lora methods, but I do not find much details for the lora training, for example, how many data samples are used, or how many steps are used for the training. It is better to provide the details to ensure that the comparison is fair.

\- In the paper, it mentions that the number of components $K$ is set to 10 which leads to  58.83% mean accuracy for Llama-7B. However, in Table 5 in the appendix, it shows that 58.83 is achieved with $K=1$. This is different with the claim for $K=10$ in the main paper. It also recommend using $K\le 3$ in Appendix B, but the main paper uses $K=10$. It is better to check this.

\- Finding 3 may have some issues. It is very weird with so many different values of K for different layers or heads. It does not seem to be a general feature. It seems that some specific K which are carefully selected by authors can lead to Figure 3. And the component are scaled by 1000. We are not sure whether 1000 is a reasonable scale number. The observation and discussion in finding 3 is heuristic and experimental. We are not sure whether it is general enough to lead to the claimed finding 3. It is better to provide more discussions.

\- In equation (8), the sum is computed over $d_h$. However, in equation (4), the sum is computed over $K$. Since their outputs refer to the same thing for the activation compensation, these two equations are not aligned and it seems to have some errors in the equations.

[R1] Fluctuation-Based Adaptive Structured Pruning for Large Language Models, AAAI 2024

[R2] EoRA: Training-free Compensation for Compressed LLM with Eigenspace Low-Rank Approximation

---

> ### Author Rebuttal · Authors · 2025-07-29
>
> Dear Reviewer aFYT,
>
> Thank you for reviewing our paper. The concerns you raised mainly focus on the **novelty, clarification of our findings, and experimental details**. We have carefully addressed all of these points, and we are glad that your suggestions have helped us improve the paper. Our responses [**R**] to your raised questions [**Q**] and weaknesses [**W**] are as follows.
>
> [**W1**] Novelty clarification.
>
> [**R1**] Thank you for your question. We **respectfully disagree** and would like to emphasize that our proposed RestoreLCC is conceptually and technically distinct from R1, R2, and their combination (R1+R2). We clarify the novelty of our method from both **theoretical and empirical** perspectives as follows.
>
> (1) Unique Insight 1 – Focusing on **a few discriminative heads (ours) rather than all modules** (R1, R2 and other studies). As shown in Section 3 (Findings 1 and 2), we reveal that reintroducing lost components to **selected pruned attention heads** can significantly restore model performance. In other words, not all modules are useful. However, previous approaches that recover all attention heads and all FFNs result in lower efficiency and often limited recovery performance.
>
> (2) Unique Insight 2 – **Recovering discriminative components instead of principal components**. Another key difference from R1, R2, and similar work is that RestoreLCC focuses on discriminative components/information rather than principal components. Previous methods emphasize recovering principal components or minimizing global losses such as MSE of hidden states. In contrast, our study (Findings 3 and Figure 3) shows that discriminative components are more effective than principal components for recovery. For example, Figure 9 shows that some minor components (indices > 20), even with extremely small eigenvalues, retain high magnitudes after tuning and contribute strongly to recovery. These components may not capture principal information, but they are highly discriminative. This phenomenon parallels the difference between PCA (capturing principal variance) and LDA (capturing discriminative directions) in classical machine learning theory. This insight also explains why we train coefficients for all components rather than only the top‑k principal components like previous studies.
>
> (3) Methodological novelty – **Leveraging the above insights and address key challenges**.While these findings reveal promising opportunities, they also **raise challenges**. For example, how to identify discriminative heads? How to find discriminative components (possibly among minor components) and determine their coefficients? To tackle these issues, we design RestoreLCC, which integrates a contrastive probing module and a lost component compensation module. Although some terms may sound similar to those in previous work, the underlying mechanisms and method operations are fundamentally different and novel.
>
> (4) **Significant improvements over previous studies**.
> We compared RestoreLCC with FLAP (R1) and EoRA (R2). As shown in the below table (same setup as lines 279–281 and Table 1), RestoreLCC outperforms EoRA by **2.95%** on Wanda-pruned models and by **3.79%** on SparseGPT-pruned models.
>
> | Wanda| PPL↓| BoolQ| RTE| Hella| Wino| ARC-e| ARC-c| OBQA| Mean|
> |-|-|--|--|---|-|--|-|--|--|
> | EoRA | 7.14 | 74.16 | 60.29 | 51.27     | 68.27 | 70.96 | 37.63 | 28.6 | 55.88 |
> | RestoreLCC| 6.93 | 72.84 | 69.68 | 56.34  | 65.98 | 71.8| 40.96 | 34.2 | 58.83 |
> | SparseGPT| PPL↓| BoolQ | RTE   | Hella| Wino|ARC-e | ARC-c | OBQA | Mean  |
> | FLAP|10.57| 68.81 | 54.15 | 44.48 | 64.4| 63.97 | 30.03 | 24 | 49.98 |
> | EoRA|9.87| 71.68 | 59.93 | 44.65  | 64.33| 63.72 | 30.38 | 23.8 | 51.21 |
> | RestoreLCC | 8.99 | 73.61 | 63.9|51.71| 65.11| 68.14 | 33.53 | 29 | 55  |
>
> [**W2**] More details of probing classifiers.
>
> [**R2**] Thank you for the question. We apologize for not providing enough details about the probing classifiers. The procedure is described in Section 4.1, and we will add the following clarification for the classifiers:
>
> Specifically, for each classifier, we use the formulation: $y = \sigma(Wm + b)$, where $m$ is either $m^+$ or $m^-$, $W$ is a weight matrix (input dimension equal to the size of $m$, output dimension = 1), $b$ is a bias term, and $\sigma$ is the sigmoid function. The output $y \in [0,1]$ indicates contradiction or entailment. We use a cross‑entropy loss optimized with Adam (lr = 1e‑2). The dataset is split into training and validation sets with a 7:3 ratio, and models are trained for 100 epochs based on empirical experience. We report probing accuracy on the validation set and **rank the importance of attention heads according to their probing accuracy**.
>
> [**W3**] Ablation study: number and effect of samples.
>
> [**R3**] We apologize for not providing enough details.
>
> For all experiments, we use the same sample sizes: general recovery – 1,000 Alpaca samples for probing and the full 52K for tuning; task‑specific recovery – 100 samples for both probing and tuning. The number of contrastive pairs equals the number of probing samples.
>
> Sample size effect: for general recovery, 1,000 samples are sufficient (accuracy ≈ 58.8); for task‑specific recovery on BoolQ, 200 samples can be sufficient (accuracy≈ 76).
>
> |General Recovery|100|200|500|1000 (reported) |3000|   |   |
> |------|-------|-------|----|---|--|------|------|
> | Mean Accuracy| 57.63 | 58.14 | 58.69 | 58.83  | 58.81 | |    |
> | BoolQ recovery| 10 | 20| 50| 100 (reported)| 200| 500 |1000|
> | Acc.| 70.03 |73.09|74.98|75.32|76.15|76.12|76.51|
>
> [**W4**] Details for baselines.
>
> [**R4**] We apologize for not providing these details. For LoRA and DoRA, we use the same settings: α = 16 and rank = 8. Regarding the applied modules, we try two configurations: (1) [v_proj, o_proj], which tunes only the head output matrices; and (2)[q_proj, k_proj, v_proj, o_proj], which tunes all head matrices. The best results are reported in the paper.
>
> For LoFiT, we experiment with 10%, 20%, and 30% of the heads and report the best results.
>
> The training samples are same for all methods. The batch size is set to 8. The learning rate is 1e-4. The max sequence length is 512. The number of training epochs is 2 for global recovery and 5 for task-specific recovery. All experiments are conducted on a single H100 GPU.
>
> [**W5**] CLarification on K.
>
> [**R5**] We apologize for the oversight. The setting of $K$ is clarified as follows, and we will double‑check similar details to ensure correctness.
>
> In our experiments, we set $K = 1$ (not 10) to obtain the reported results.
> We recommend using $K \leq 3$ because performance is consistently better in this range than with $K > 3$. Moreover, after the top 3 components, eigenvalues become very small, so additional components have negligible impact on the final results while increasing computation. Therefore, using $K \leq 3$ provides both better performance and efficiency.
>
> [**W6**] Clarification on Finding 3.
>
> [**R6**] Thank you for the question. We clarify Finding 3 as follows.
>
> Unlike previous studies that assume principal components are always useful for recovery, we present a different insight: **discriminative information can also reside in minor components rather than principal components**. Principal components may capture high-variance but task‑irrelevant information, while minor components may encode task‑specific signals. This distinction is analogous to PCA vs. LDA in classical machine learning.
>
> To verify this, we manually inspected the token distributions of each component (Eq. 6). For example, in BoolQ, the top tokens of some principal components (e.g., 1st or 2nd) may include stopwords like *a*, *the*, or unrelated words like *walk*. In contrast, **some minor components have answer tokens like *yes* and *no*, which are highly relevant to the task**. Such task‑specific minor components are more discriminative and recover performance more effectively. We further validated this hypothesis by manually selecting attention heads and comparing recovery using principal vs. discriminative components. Figure 3 illustrates this phenomenon. Figure 9 also validates this: **the trained magnitudes of minor components can become larger than those of principal components due to their discriminative power**.
>
> Scaling factor: Eigenvalues of principal components can be more than 1000× larger than those of minor components. To ensure fair comparison, we apply a scaling factor (1000) when adding minor components so their contributions are comparable.
>
> In summary, the key finding is that task‑specific discriminative information may reside in minor components, not just in principal components. This observation directly motivates RestoreLCC, whose goal is to leverage these discriminative components for effective recovery.
>
> [**W7**] Clarification on Eq. 4 and Eq.8.
>
> [**R7**] Thank you for the question. We would like to clarify that there is no error in either equation. In Eq. 4, the top‑k components are used only for probing (Section 4.1), whereas in Eq. 8 all components are used for recovery tuning.
>
> This is because, in the first step, we use only the top‑k components to evaluate the discriminative power of each head, as minor components have extremely small eigenvalues and barely affect probing results. However, as highlighted in Finding 3, during recovery for the selected heads, **important discriminative information may also lie in minor components rather than only in the top‑k ones**.
>
> Therefore, our method considers all components and learns a magnitude for each component (total of $d_h$) so that even minor but discriminative components can receive a large weight for recovery. Figure 9 also confirms this behavior. Therefore, in Eq. 8 it should indeed be $d_h$.
>
> **Thank you once again for reviewing our paper. We will incorporate the clarifications and discussions into the revised version. Please let us know if you have any further feedback, and we look forward to your reply.**

---

> > ### Comment · Reviewer_aFYT · 2025-08-04
> > **discussion**
> >
> > Thanks for the rebuttal. Most of my concerns are addressed and I will update my score.

---

> ### Author Response · Authors · 2025-08-05
>
> Dear Reviewer aFYT,
>
> Thank you very much for taking the time to review our rebuttal and for re-evaluating our paper. We are pleased to hear that most of your concerns, such as those related to the novelty and experimental aspects, have been satisfactorily addressed. We will thoughtfully incorporate all relevant points from the discussion into our revised manuscript.
>
> Best regards,
>
> Authors of Paper 7969

---

### Official Review · Reviewer_oP8x · 2025-07-02

**Clarity:** 3
**Significance:** 3
**Originality:** 3
**Rating:** 5
**Confidence:** 4

**Summary:**

Network pruning reduces the complexity of a trained model, enabling more efficient deployment, but at the cost of potentially reduced quality.  Existing approaches to efficiently recovering quality (i.e., excluding fine-tuning the full network) do not consider the pruning that occurred when performing recovery.  The authors perform an empirical study to gain insight into how leveraging the information in *lost* activations can be used to restore quality.  Using these insights, RestoreLCC injects the lost information back into the model via a learned bias term in a three step process: (1) contrastive probing identifies the important attention heads, (2) Lost Component Compensation (LCC) performs SVD on the difference between the outputs of the important dense and pruned heads and learns optimal magnitudes for the top components before (3) augmenting the important heads with these learned bias terms, "re-injecting" the lost information back into the model.  Empirical studies show improvements on most metrics compared to baselines.

**Questions:**

1. What weight tensors are pruned in your experiments?  My understanding of the submission is that it's only Q, K, V, and dense output projections in attention blocks, not MLP blocks. (Clarity, Significance)
2. Is there anything fundamentally stopping RestoreLLC from being used on heavily quantized models, or does it only apply to pruned models? (Significance)
3. How does EoRA compare for some model, given the same sparsities and tasks? (Quality, Significance)
4. What software was used, and how much computation time was required to probe and learn the compensatory terms for each model? (Quality)

**Ethical Concerns:**

["NO or VERY MINOR ethics concerns only"]

**Final Justification:**

All of my questions were resolved with the authors' response.  Assuming the authors can address the clarity issues raised by myself and the other reviewers, I have no other issues.

**Limitations:**

This approach seems to be limited to attention weights, and cannot address MLPs.

**Quality:**

3

**Strengths And Weaknesses:**

## Strengths
The problem statement is well-motivated, and the approach seems novel, is theoretically-grounded, and is generally successful.  The solution's low-overhead is particularly compelling; additional bias terms are a very small tax on model size and deployment efficiency, in contrast to the LoRA family of techniques, which require additional (though relatively small) weight matrices.  The contrastive probing and compensation process is adequately described such that an expert could reproduce the results.  The evaluations are sufficiently broad with respect to models and tasks to inspire confidence that there are no surprises hidden behind under-explored areas.  This particular solution to the problem incorporates existing ideas, but is largely original.

## Weaknesses
Clarity is a mixed bag - while the exposition is clear, some details are not.  For example, the precise sparsity imposed is unclear.  At times the activations are pruned (line 113), but later it is the attention heads (I assume weights) that are pruned (line 132).  It's also not clear which weights are pruned; is it only those in the attention blocks Q, K, V and dense output projections, or also weights in the MLP blocks (which can be much larger due to the expansion factor)?  This has direct bearing on the significance of the submission.  Also, consider avoiding the term "performance;" it is overloaded and can mean both model quality as well as execution speed or efficiency.  In this context, where both meanings are plausible, it's better just to be explicit.

Quality is also slightly lacking.
- The inference speed overheads in Appendix C isn't very useful without knowing the deployment conditions: hardware, software, type of pruning, batch sizes, sequence lengths, etc. all play a large role in both dense and sparse efficiency.  Further, the magnitude of the speedups will influence how important the overheads are.
- The ablation study comparing identifying important heads using contrastive probing vs. randomly-selected heads.  Surely there's a simpler metric that could perform better than random choice: MSE between the output of dense and pruned heads, K-L Divergence, etc.
- Figure 5 shows the top-5 decoded tokens, but the text suggests it shows the top-10 (line 347)
- There are two references that should be considered for discussion and inclusion:
    - Moore and Chaudhuri probe networks' activations in order to drive pruning decisions directly.
    - Liu et al. do take the pruning into account when restoring quality with low-rank adaptation.  This work, in particular, should be included as a baseline, since it stands in contrast to, as claimed in the abstract, "most PEFT methods are designed for dense models and overlook the distinct properties of pruned models."


1. Moore and Chaudhuri, Using noise to probe recurrent neural network structure and prune synapses, NeurIPS 2020
2. Liu et al., EoRA: Fine-tuning-free Compensation for Compressed LLM with Eigenspace Low-Rank Approximation, https://arxiv.org/abs/2410.21271

---

> ### Author Rebuttal · Authors · 2025-07-29
>
> Dear Reviewer oP8x,
>
> Thank you for taking the time to review our paper and for recognizing it as **well‑motivated, novel, theoretically grounded, and successful**. The concerns you raised mainly focus on **comparisons with different head‑selection methods, EoRA, extension to quantized models, and clarifications on pruning**. We have carefully addressed all of your concerns, and we are glad to see that, with your suggestions, the quality of the paper has been greatly improved. Our responses [**R**] to the raised weaknesses [**W**] and questions [**Q**] are summarized as follows.
>
> [**W1**] Clarify pruning-related terms
>
> [**A1**] Thank you for pointing this out. We apologize for not explaining these terms clearly. Below we clarify the terminology:
>
> * LLM pruning:
> Current LLM pruning methods focus on pruning all the weight matrices (including attention matrices $Q, K, V, O$ and MLP block matrices) rather than the activations of each Transformer block. For example, Wanda removes 50% of the weights in each matrix of both the attention and MLP modules (setting them to zero), leading to an overall sparsity of 50%. Other methods follow a similar manner. **Therefore, in our paper, all weight matrices in the LLM are pruned, and we focus on restoring them by compensating through attention heads. (The reasons for not using MLPs/FFNs are explained in our response R10 below.)**
>
> * Activation:
>   An activation refers to the output of a specific module.
>
> * Pruned activation: It is the activation produced by a pruned module. While the activations have the same dimension as those from the original model, they may be less informative because the underlying weight matrices are pruned. Note that **the activations themselves are not pruned**; only the weight matrices are. We acknowledge that the term “pruned activation” may be confusing, and will revise this terminology for clarity.
>
> * Sparsity: It refers to the proportion of weight parameters that are pruned. The sparsity settings for our experiments can be found in Section 5. For Wanda and SparseGPT, each weight matrix is pruned with the same ratio (e.g., 50% parameters removed), resulting in an overall sparsity of 50%. For SlimGPT, a dynamic ratio is applied to different layers (e.g., 12.5% in shallow layers and 30% in deeper layers, giving an overall sparsity of 20%) based on a specific ratio-assignment formula.
>
> * Performance: Based on your suggestions, we will explicitly use terms such as quality, execution speed, or efficiency in different contexts when discussing performance.
>
>  As the report results in Tables 1 - 3, our proposed method can be applied to all three different pruning paradigms.
>
> [**W2**] Overhead details in Appex. C.
>
> [**R2**] We apologize the for the missed details. For LoRA and DoRA, we use the same settings: α = 16 and rank = 8. Regarding the applied modules, we try two configurations: (1) [v_proj, o_proj], which tunes only the head output matrices; and (2)[q_proj, k_proj, v_proj, o_proj]`, which tunes all head matrices. The best results are reported in the paper.
>
> For LoFiT, we experiment with 10% - 30% of the heads and report the best results.
>
> The batch size is set to 8. The max sequence length is 512. All the experiments are conducted on a single H100 GPU.
>
> The overhead is reported using the best hyperparameter settings, with the base model being LLaMA‑7B pruned by Wanda at 50% sparsity. The inference speed is measured as the ratio of the total testing time for general recovery (lines 279–281) of the restored model to that of the original pruned model.
>
> [**W3**] Compare with other metric-selected heads.
>
> [**R3**] Thank you for your suggestion. We have conducted additional experiments with two alternative head-selection strategies: (1) MSE-selected heads: selecting heads with the smallest MSE between the outputs of the dense and pruned models. (2) KL-selected heads: selecting heads with the smallest KL divergence. The results are as follows: our probing-based selection consistently identifies important heads and is more effective than other metric-based approaches.
>
> |Method| Accuracy |
> |-------|---|
> |RestoreLCC | 58.83|
> |random| 57.57|
> |MSE-selected| 58.14|
> |KL-selected| 57.92|
>
> [**W4**] line 347: top-10 should be top-5.
>
> [**R4**] We apologize for the typo. We will correct “top‑10” to “top‑5” and carefully double‑check the paper for similar issues.
>
> [**W5**] Discuss two references.
>
> [**R5**] Thank you for you suggestions. We have carefully read both references and will add to our revised paper.
>
> (1) (add to related work) Moore and Chaudhuri utilize activation noise to probe network structure and identify redundant neurons. (add to future work) They also discuss a novel way of leveraging activation noise for neuron identification. We believe activation noise could similarly be used to enhance contrastive probing, and we plan to explore this direction in future work.
>
> (2) EoRA. (add to related work) EoRA provides a fine-tuning-free approach to recover pruned models by searching low-rank spaces in a task-specific eigenspace to minimize compression loss.
> (add to Table 1) We also compare our method with EoRA, and the results are summarized below (Detailed results can be found in R8 below). It can be observed that our method significantly outperforms EoRA.
>
> |Wanda| Mean  |
> |------|----|
> | EoRA |55.88|
> | RestoreLCC|58.83|
> | SparseGPT| Mean|
> | EoRA|51.21|
> | RestoreLCC|55|
>
> [**Q1**] What weight tensors are pruned?
>
> [**R6**] In our experiments, all weight matrices in both the attention and MLP blocks are pruned, following the exact pruning settings of the respective methods: Wanda, SparseGPT, and SlimGPT.
>
> These three approaches are well‑established state‑of‑the‑art pruning methods, each representing a different pruning paradigm. The primary difference lies in the positional constraints of pruned parameters in a matrix. To demonstrate the generalizability of our method across different pruning strategies, we evaluate RestoreLCC on all three types of pruned models, as shown in Tables 1–3.
>
> Our recovery operations focus on the attention modules in order to restore the performance of the entire model. We also discuss the possibility of recovering MLPs (FFNs) in our response [**R10**] below.
>
> [**Q2**] Application to heavily quantized models.
>
> [**R7**] Thank you for the question. Our method can also be applied to heavily quantized models. We conducted experiments with 4‑bit quantization on the pruned model (by Wanda) and report the recovery performance below. The experimental setup follows lines 282–285 and Table 3 in our paper. As shown, the proposed RestoreLCC successfully recovers the performance of heavily quantized LLMs.
>
> |data| BoolQ| RTE| ARC-e| ARC-c |
> |--------|-------|-----|-------|-------|
> | 4-bit quantized model|68.2|57.76| 59.89| 29.95|
> | RestoreLCC (Ours)| 72.97|68.23|61.03|33.02|
>
> [**Q3**] Comparison with EoRA.
>
> [**R8**] Thank you for the question. We compare our method with EoRA, and the results are summarized in the table below. It can be observed that our method significantly outperforms EoRA. The experiment setup follows lines 279-281 and Table 1 in our paper.
>
> |Wanda| PPL↓ | BoolQ | RTE| Hella| Wino| ARC-e | ARC-c|OBQA|Mean|
> |--------|------|----|-------|-----|------|----|-------|------|-------|
> | EoRA |7.14|74.16|60.29| 51.27| 68.27| 70.96| 37.63| 28.6| 55.88|
> | RestoreLCC (Ours) | 6.93 | 72.84 | 69.68|56.34| 65.98|71.8|40.96| 34.2| 58.83|
> | SparseGPT  | PPL↓ | BoolQ | RTE   | Hella| Wino| ARC-e | ARC-c | OBQA | Mean|
> | EoRA | 9.87|71.68|59.93|44.65|64.33|63.72|30.38| 23.8 |51.21|
> | RestoreLCC (Ours)| 8.99 |73.61| 63.9| 51.71| 65.11|68.14|33.53|29|55|
>
> [**Q4**] The used software and other overhead.
>
> [**R9**] Thank you for your question. We report the details of recovering pruned LLaMA‑7B (1 H100 GPU, torch.bfloat16 precision, max_length=512, batch size=8, Alpaca Dataset).
>
> The **software** we used is Python and torch. All the required packages and required versions can be found in our submitted materials: requirements.txt.
>
> The **computation time for probing** is 8min23sec.
>
> The **1-epoch training time for each method** is as follows. For training time, the methods rank as follows:
> RestoreLCC < LoRA < DoRA < LoFiT. For GPU memory usage: LoRA < RestoreLCC = LoFiT < DoRA. Compared with the DoRA and LoFiT, RestoreLCC has the lowest overall overhead while providing superior performance.
>
> | llama-7B|Time|Memory|
> |-------|-----|-------|
> | LoRA| 4h20min| 61GB|
> | DoRA | 6h08min| 71GB|
> | LoFiT| 8h11min| 65GB|
> | RestoreLCC| 4h13min|65GB|
>
>  [**Q5**] Limitations of addressing MLPs.
>
>  [**R10**] Thank you for raising this concern. **Our goal is to restore pruned models in which all weight matrices in both the attention and MLP modules have been pruned**. To achieve this, our recovery operations focus on the attention modules rather than the MLPs to restore the overall model performance.  We provide both theoretical and empirical analyses to support this choice.
>
>  (1) Studies on mechanistic interpretability show that **attention heads specialize in distinct functions for different tasks [1,2,3], while FFNs mainly store knowledge and map inputs to outputs [4,5]**. The detailed discussion and references can be found in our response **R1** to Reviewer Kk22.
>
> (2) Empirical evidence. We would like to highlight that **our method can be directly applied to FFNs**. We select heads instead of FFNs for recovery also because of the empirical results. Compensating attention achieves 58.83, while compensating FFNs only reaches 57.00.
>
> |    | BoolQ| RTE| Hell. | Wino. | ARC-e | ARC-c | OBQA| Mean|
> |------|-----|-----|------|----|-----|-----|-----|------|
> | Recover attn. | 72.84 | 69.68 | 56.34| 65.98 | 71.8 | 40.96 | 34.2 | 58.83|
> | Recover FFNs | 69.42 | 63.54 | 55.54| 67.17|70.96|39.59|32.8| 57|
>
> **Thank you again for the insightful comments. We will incorporate the clarifications and discussions into the revised version and we look forward to your feedback.**

---

> > ### Comment · Reviewer_oP8x · 2025-08-04
> >
> > I thank the authors for their in-depth and informative responses to all reviewers.  I find that my major concerns were due to my misunderstanding the scope of the submission - since it *does* apply sparsity to MLP weights as well, only using recovery on the attention weights, the significance of the work is greatly increased.  I'd urge the authors to make this clear so that future readers do not suffer the same conclusion that I did.  I trust that this, as well as the clarity issues raised by other reviewers, can be addressed in a future version of the paper.
> >
> > With this knowledge and the authors' new data adequately addressing my other questions, I'm pleased to raise my score to "accept."

---

> ### Author Response · Authors · 2025-08-04
>
> Dear Reviewer oP8x,
>
> Thank you very much for taking the time to carefully review our rebuttal responses. We truly appreciate your thoughtful feedback. We will ensure that this point, along with the other comments raised in the rebuttal, are clearly addressed and reflected in the revised manuscript.
>
> Best regards,
>
> Authors of Paper 7969

---

### Official Review · Reviewer_n2VC · 2025-07-03

**Clarity:** 3
**Significance:** 3
**Originality:** 3
**Rating:** 5
**Confidence:** 4

**Summary:**

This paper introduces RestoreLCC, a novel and targeted restoration strategy for pruned large language models (LLMs). Unlike prior parameter-efficient fine-tuning (PEFT) methods that overlook the unique properties of pruned models, RestoreLCC explicitly identifies and compensates for the lost information caused by pruning. The method consists of two key components: (1) Contrastive Probing, which identifies important attention heads via contrastive activation editing; and (2) Lost Component Compensation (LCC), which reconstructs missing activation components using learned linear combinations of principal directions and injects them back to recover the model’s capability. Extensive experiments across multiple pruning schemes (structured, semi-structured, unstructured) and LLM variants demonstrate that RestoreLCC achieves superior accuracy and perplexity recovery without compromising sparsity or inference efficiency.

**Questions:**

**Scope of Probing Generalization**: Given that contrastive probing is data/model/pruning-specific, how scalable is RestoreLCC across multiple downstream tasks without retraining the probing module?

**Efficiency at Scale**: While parameter overhead is small, does RestoreLCC scale gracefully with larger models (e.g., >30B)? Are there latency/throughput benchmarks in real-world scenarios?

**Ethical Concerns:**

["NO or VERY MINOR ethics concerns only"]

**Final Justification:**

## Final Justification

I have raised my score to **Accept**. The authors provided an exceptionally strong and thorough rebuttal that has fully addressed all of my initial concerns. The new experiments and clarifications have significantly strengthened the paper's claims and overall contribution.

---

### Resolved Issues

**Model Generalizability**
My primary concern regarding the method's generalizability being limited to LLaMA-family models was convincingly resolved. The authors added extensive experiments on entirely different and recent architectures (**Qwen3**, **DeepSeek-R1-Qwen3**), demonstrating that **RestoreLCC** consistently outperforms baselines. This provides compelling evidence for the method's broader applicability.

**Practicality and Scalability**
My questions about the practical deployment, specifically the portability of the probing module and the method's efficiency at scale, were also fully addressed with new, targeted experiments. The authors demonstrated:

- Successful cross-task performance recovery without retraining the probing module, directly answering my scalability question.
- Effective performance on a much larger **70B model (LLaMA-70B)**, proving scalability.
- A concrete latency benchmark showing a negligible inference overhead of ~**3%**, confirming real-world efficiency.

---

### Unresolved Issues

From my perspective, there are **no remaining unresolved issues**. The authors' response was comprehensive and satisfying.

---

### Overall Assessment

The weight of these resolved issues is high, as they were central to validating the paper's core claims of being a **generalizable** and **practical** method. The authors' diligence in running these new experiments has transformed the paper from a promising one with limitations to a **robust and well-supported contribution**.

---

### Key Points for the Authors

- Your rebuttal was exemplary. The new experiments were precisely what was needed to validate your claims and were highly persuasive.
- The paper is now substantially stronger, with robust evidence backing the **generalizability**, **cross-task portability**, and **efficiency** of RestoreLCC.
- Please ensure that all the new results and the insightful discussions from the rebuttal are **carefully and clearly integrated** into the final version of the manuscript.

**Limitations:**

Yes, but not enough.

**Paper Formatting Concerns:**

Lines 38–59 in the main body contain a dense comparison of Dense, Pruned, LoRA, and LCC variants with overlapping figure references. The readability could be improved by better paragraphing or figure placement.

Figure labels (e.g., “l12.a16 l5.a20”) could be clarified with axis legends or tooltips for non-expert readers.

**Quality:**

3

**Strengths And Weaknesses:**

### Strengths

- The paper introduces a unique insight—model degradation from pruning is traceable to lost activation components—which leads to a targeted and explainable restoration method.

- The methodology is well-justified with theoretical and empirical analyses. Ablation studies, probing diagnostics, and interpretability results are thorough and convincing.

- The overall exposition is clear, well-structured, and supported with informative figures (e.g., logit gain visualizations, component effects).

- The work addresses a critical bottleneck in deploying sparse LLMs by offering a low-overhead, generalizable recovery mechanism. It shows consistent gains over strong PEFT baselines like LoRA, DoRA, and LoFiT.

### Weaknesses

- The scope of evaluation is currently limited to LLaMA-family models. Including more diverse architectures (e.g., Mistral, Qwen, Baichuan) would strengthen the claim of generalizability.

- The contrastive probing module depends on data/model-specific classifier training, which could affect practical deployment and cross-task portability.

- A few sections (e.g., lines 38–59) in the main text are densely written, with overlapping figure references, which slightly affects clarity.

---

> ### Author Rebuttal · Authors · 2025-07-29
>
> Dear Reviewer n2VC,
>
> Thank you for taking the time to review our paper and for recognizing it as **novel, with unique insights that are clearly and extensively justified**. The concerns you raised mainly relate to **extending our approach to other LLMs, scalability and the portability of the probing module**. Following your suggestions, we have added experiments and additional analysis to address these points. We are glad to see that, with your feedback, the paper has been greatly improved. Our responses [**R**] to your mentioned weaknesses [**W**] and questions [**Q**] are summarized below.
>
> [**W1**] Evaluate on more LLMs from different families.
>
> [**R1**] Thank you for raising this question. To verify the generalizability of RestoreLCC on more recent LLMs, we evaluated it on the latest **Qwen3 family and DeepSeek‑R1‑Qwen3‑8B**, both released at the end of **May 2025**. Our results on Qwen3‑8B, Qwen3‑14B, and DS‑R1‑8B are as follows. Compared with other baseline methods, RestoreLCC consistently delivers significant improvements.
>
> We would also like to highlight an interesting observation on the latest Qwen3 models: the recovered model performs even better than the original dense model. We believe this is because, despite the state‑of‑the‑art performance of Qwen3, the model still benefits substantially from additional tuning with the Alpaca instruction‑tuning dataset.
>
> | Qwen3-8B | BoolQ | RTE   | HellaSwag | WinoGrande | ARC-e | ARC-c | OBQA  | Mean  |
> |----------|-------|-------|-----------|------------|-------|-------|-------|-------|
> | Dense    | 86.57 | 78.34 | 57.10     | 67.80      | 83.54 | 55.80 | 31.00 | 65.74 |
> | Wanda    | 84.86 | 70.04 | 50.12     | 69.46      | 80.22 | 50.85 | 28.20 | 61.96 |
> | DoRA     | 86.30 | 77.26 | 56.13     | 68.82      | 82.37 | 53.58 | 31.80 | 65.18 |
> | Ours     | 87.13 | 79.06 | 57.29     | 70.48      | 84.68 | 57.59 | 32.60 | **66.98** |
>
>
> | Qwen3-14B | BoolQ | RTE   | HellaSwag | WinoGrande | ARC-e | ARC-c | OBQA  | Mean  |
> |-----------|-------|-------|-----------|------------|-------|-------|-------|-------|
> | Dense     | 89.33 | 77.62 | 60.97     | 73.01      | 84.22 | 58.62 | 35.00 | 68.40 |
> | Wanda     | 87.43 | 72.92 | 57.69     | 69.77      | 83.50 | 55.97 | 33.60 | 65.84 |
> | DoRA      | 88.07 | 78.70 | 59.87     | 74.66      | 84.09 | 57.85 | 34.50 | 68.25 |
> | Ours      | 89.57 | 80.14 | 60.70     | 74.74      | 86.15 | 60.15 | 35.00 | **69.49** |
>
> | DS-R1-8B | BoolQ | RTE   | HellaSwag | WinoGrande | ARC-e | ARC-c | OBQA  | Mean  |
> |----------|-------|-------|-----------|------------|-------|-------|-------|-------|
> | Dense    | 85.87 | 80.51 | 58.54     | 67.01      | 80.18 | 51.71 | 31.60 | 65.06 |
> | Wanda    | 82.48 | 77.26 | 51.53     | 67.17      | 75.67 | 48.12 | 28.40 | 61.52 |
> | DORA     | 84.80 | 74.37 | 56.84     | 68.35      | 77.90 | 50.09 | 32.30 | 63.52 |
> | ours     | 87.22 | 74.01 | 57.41     | 69.46      | 83.84 | 55.63 | 33.80 | **65.91** |
>
>
> [**W2**] The contrastive probing module depends on data/model-specific classifier training, which could affect practical deployment and cross-task portability.
>
> [**R2**] Thank you for pointing out this concern. We would like to highlight that (1) the overhead of training the classifiers on new models and datasets is negligible, (2) the probing classifier is very easy to implement for new datasets and new models, (3) cross‑task portability can be addressed using the Alpaca dataset when no data is available for a new task, and (4) **the benefits of probing far outweigh its cost since it avoids tuning all heads**. We agree this is an interesting direction and plan to explore task‑agnostic probing strategies in future work to further improve portability. We further clarify these four points as follows:
>
> (1) Negligible overhead. For each attention head, we train only a simple linear layer as the probing classifier. This process is precomputed before the recovery tuning stage (Section 4.2 LCC). It can be completed even on a low‑end GPU (e.g., NVIDIA RTX 1080), takes only a few minutes, and requires very little GPU memory. Therefore, the overhead is negligible.
>
> (2) Easy implementation. For new datasets and models, the only step needed is to obtain head‑wise activations from the new dataset and model. The code implementation is simple and we provide three different ways:
> (i) Use Python “hook” functions to capture the inputs/outputs of attention modules directly.
> (ii) Use the open‑source Pyvene package, which provides convenient APIs to extract activations.
> (iii) Use the open‑source TransformerLens package, which supports easy activation extraction even for the latest models (e.g., Qwen3).
> In all three cases, only a few lines of Python code are needed, after which the classifiers can be quickly trained.
>
> (3) Cross‑task portability. When there is no dataset for a new task, the Alpaca instruction‑tuning dataset can be used in a general recovery setting (lines 279‑281). As shown in Table 1, this dataset works well for both language modeling and seven commonsense reasoning tasks, enabling recovery without task‑specific data. Therefore, in such cases, **the Alpaca dataset can serve as a universal fallback for probing and tuning across different datasets**. This substantially improves the portability of our method.
>
> (4) **Benefit outweights cost**. Probing allows us to focus on informative heads: in practice, only 10%–30% of heads are restored. This greatly improves efficiency by avoiding unnecessary tuning of all heads. Hence, the efficiency gains from probing far outweigh its small cost.
>
>
> [**W3**] A few sections (e.g., lines 38–59) in the main text are densely written, with overlapping figure references, which slightly affects clarity.
>
> [**R3**] Thank you for pointing this out, and we apologize for the inconvenience caused. We will carefully double‑check and revise all such presentations to ensure clarity and correctness.
>
> [**Q1**] Scope of Probing Generalization: Given that contrastive probing is data/model/pruning-specific, how scalable is RestoreLCC across multiple downstream tasks without retraining the probing module?
>
> [**R4**] Thank you for your question. We would like to clarify that **the probing module generalizes well across downstream tasks without requiring retraining**. This is supported by the following two observations:
>
> (1) Probing on a general dataset, then testing on diverse downstream tasks. Our general recovery setting (lines 279–281) follows this requirement. For the results in Table 1, attention heads are probed using the Alpaca instruction‑tuning dataset, while the evaluation tasks are very different, including a language modeling dataset and seven commonsense reasoning benchmarks. As shown in Table 1, heads probed on general instructions successfully recover performance on diverse tasks such as BoolQ and ARC.
>
> (2) Probing on one specific downstream dataset, then testing on other tasks. To further verify the scalability, we use **probing results obtained on the BoolQ dataset and apply them for recovery on other datasets including RTE, ARC‑e, and ARC‑c**. The below table shows the results. Even without retraining, the pruned model’s performance is successfully recovered on these tasks, achieving better results than the pruned baseline. It is notable that the results of original pruned model are from Table 3 in our paper.
>
> | Tasks                                 | RTE   | ARC-e | ARC-c |
> |---------------------------------------|-------|-------|-------|
> | Pruned Model                          | 59.21 | 62.67 | 30.29 |
> | RestoreLCC without retraining probing | 68.23 | 62.92 | 35.41 |
>
> [**Q2**] Efficiency at Scale: While parameter overhead is small, does RestoreLCC scale gracefully with larger models (e.g., >30B)? Are there latency/throughput benchmarks in real-world scenarios?
>
> [**R5**] Thank you for your question.
>
> Firstly, RestoreLCC scales well to larger models. We conducted task‑specific recovery (lines 282–285 in our paper) on the BoolQ dataset using LLaMA‑70B. The results on the below table show that our proposed RestoreLCC continues to outperform the LoRA baseline.
>
> Due to the significant time and GPU resources required for tuning a 70B model, we were only able to complete the current experiment before July 30, 2025. We will continue this work and include additional results in the revised version of the paper.
>
> | llama-70B    | BoolQ |
> |--------------|-------|
> | Pruned Model | 84.7  |
> | LoRA         | 86.21 |
> | RestoreLCC   | 87.25 |
>
>
> Secondly, regarding latency and throughput, we measured the inference speed on the BoolQ dataset after recovery. The delay is calculated as the ratio of total inference time with RestoreLCC to that of the original pruned model. This ratio is approximately 1.03, indicating only a ~3% slowdown, which can be ignored. Thus, RestoreLCC maintains nearly the same inference speed as the original pruned model while delivering significantly better performance.
>
> [**Q3**] Concerns regarding the figure placement and labels.
>
> [**R6**]  Thank you for pointing out these concerns. We will carefully review all figures and make the necessary adjustments to their placement, labels, and formatting to ensure that they are clear and easy to read.
>
>
> **Thank you once again for reviewing our paper and for your insightful comments. We will incorporate all of these discussions into the revised version. Please let us know if you have any further questions, and we look forward to your feedback.**

---

> > ### Comment · Reviewer_n2VC · 2025-08-05
> > **Official Comment by Reviewer n2VC**
> >
> > Thanks for their thorough and detailed rebuttal. The response has convincingly addressed all of my major concerns regarding model generalizability, probing portability, and efficiency at scale. The newly added experiments on diverse architectures like Qwen, the scalability test on the 70B model, and especially the cross-task generalization experiment without retraining the probing module, have substantially strengthened the paper's claims and effectively resolved my initial reservations. Consequently, I will be raising my score accordingly and I look forward to seeing these valuable additions incorporated into the revised manuscript.

---

> ### Author Response · Authors · 2025-08-05
>
> Dear Reviewer n2VC,
>
> Thank you very much for taking the time to carefully review our rebuttal. We truly appreciate your comments, and we will incorporate all these discussions into our revised paper.
>
>
> Best regards,
>
> Authors of Paper 7969

---

### Official Review · Reviewer_Kk22 · 2025-07-03

**Clarity:** 2
**Significance:** 2
**Originality:** 2
**Rating:** 4
**Confidence:** 3

**Summary:**

This paper proposes RestoreLCC, a parameter-efficient fine-tuning method designed for post-pruning model recovery. Under several different pruning settings, RestoreLCC demonstrates superior recovery performance compared to directly applying other PEFT methods.

**Questions:**

1. Could you explain why, in Figure 1, the accuracy on the right increases beyond that of the dense model after applying LCC?

2. This paper seems to lack a comparison with full-parameter fine-tuning for performance recovery.

3. I would like to know the effect of applying the method to the FFN layers.

4. Please clarify where the PEFT methods are added in the model and provide other relevant implementation details (e.g., rank, alpha, learning rate...).

**Ethical Concerns:**

["NO or VERY MINOR ethics concerns only"]

**Final Justification:**

I think the authors have partially addressed my concerns. This method can be applied in scenarios involving pruning recovery and represents a PEFT approach focused on specific contexts. I believe that after incorporating the additional content provided by the authors, this paper can be accepted.

**Limitations:**

Yes

**Quality:**

3

**Strengths And Weaknesses:**

Strengths

1. An approach similar to distillation is used to preserve errors in the vectors, which reduces the attention information loss after pruning and thus mitigates the performance degradation caused by pruning.

2. The method is validated under different pruning settings.

Weaknesses

1. The underlying rationale behind the effectiveness of the proposed method seems to lack a deeper explanation. I am curious why the information compensation is specifically applied at the attention module, rather than after the attention or at the intermediate dimensions of the FFN, for example.

2. The method appears to require using a large number of intermediate activations during recovery fine-tuning in order to compute the error vectors. I think this seems to deviate from the original purpose of using PEFT for recovery fine-tuning. Could you provide a comparison of the training overhead of different methods during the recovery fine-tuning stage? In my opinion, the number of trainable parameters is not the main concern in this stage.

3. All compared PEFT methods are limited to very small trainable parameter settings. I believe this setup is favorable to the method. However, it is clear that the training overhead of PEFT is not sensitive to the number of trainable parameters; for example, the overhead difference between rank=1 and rank=16 is minimal. I think that in the recovery fine-tuning stage, performance recovery should be the main focus, and a higher rank may yield better performance with only negligible increases in fine-tuning overhead.

---

> ### Author Rebuttal · Authors · 2025-07-29
>
> Dear Reviewer Kk22,
>
> Thank you very much for reviewing our paper and recognizing our method as **effective**. Your concerns mainly focus on the effect of RestoreLCC on **FFNs, its overhead, and clarification of certain details**.  Following your suggestions, we have added further analysis and additional experiments on these points. We are pleased that, with your feedback, the paper has been significantly improved. Our responses (**R**) to your weaknesses (**W**) and questions (**Q**) are as follows.
>
> [**W1**] Explain why RestoreLCC uses attention rather than FFN.
>
> [**R1**] Thank you for this valuable comment. The output of each Transformer block consists of the attention module output (Eq.1 in our paper) and the FFN module output. In our study, we focus on the **attention output matrix** rather than the FFN output matrix for the following reasons:
>
> (1) Studies on mechanistic interpretability show that **attention heads specialize in distinct functions for different tasks, while FFNs mainly store knowledge and map inputs to outputs**. For example, [1] finds that in arithmetic tasks, attention heads process key information and FFNs then produce the final answer. They also show that fine‑tuning only important heads outperforms tuning all parameters (including FFNs). Similarly, [2] demonstrates that different heads play distinct roles, and [3] also supports this observation. By contrast, FFNs store factual knowledge [4] and refine output logits [5]. Hence, compensating attention heads is more appropriate than compensating FFNs.
>
> (2) Attention heads provide **finer information**, whereas FFNs produce high‑dimensional, aggregated outputs. In LLaMA-7B, each head outputs 128 dimensions (32 heads per layer), while an FFN outputs 4096. Smaller head outputs allow finer analysis. Moreover, choosing 32 heads can span multiple layers, while the same size in FFN covers only one layer.
>
> (3) **Empirical evidence**. We applied RestoredLCC to FFNs and report the best results as follows (More results can be found in below reponse[**R6**].) Compensating attention achieves 58.83, while compensating FFNs only reaches 57.00.
>
> |  | BoolQ | RTE   | Hell. | Wino. | ARC-e | ARC-c | OBQA | Mean  |
> |--|---|---|---|--|----|--|--|---|
> | On attn. | 72.84 | 69.68 | 56.34 | 65.98| 71.8 | 40.96 | 34.2 | 58.83 |
> | On FFNs | 69.42 | 63.54 | 55.54| 67.17 | 70.96 | 39.59 | 32.8 | 57 |
>
> [1] Interpreting and improving large language models in arithmetic calculation, ICML 2024
>
> [2] Model Tells You What to Discard: Adaptive KV Cache Compression for LLMs, ICLR 2025
>
> [3] Dressing up llm: Efficient stylized question-answering via style subspace editing. ICLR 2025
>
> [4] Locating and Editing Factual Associations in GPT, NeurIPS 2022
>
> [5] Transformer Feed-Forward Layers Are Key-Value Memories, EMNLP 2021.
>
> [**A2**] Training overhead.
>
> [**R2**] Thank you for the question. We would like to clarify that **the overhead of our proposed RestoreLCC is smaller than advanced PEFT methods while achieving much better performance**. While recovering pruned LLaMA‑7B (1 H100 GPU, torch.bfloat16 precision, 1 epoch, max_length=512, batch size=8, Alpaca Dataset),the empirical and theoretical evidence are:
>
> (1) Empirical Analysis. The table below compares training time and GPU memory usage. For training time, the methods rank as follows: RestoreLCC < LoRA < DoRA < LoFiT. For GPU memory usage: LoRA < RestoreLCC = LoFiT < DoRA. Compared with the DoRA and LoFiT, RestoreLCC has the lowest overall overhead while providing superior performance.
>
> |llama-7B|Time|Memory |
> |----|----|---|
> | LoRA | 4h20min | 61GB |
> | DoRA| 6h08min| 71GB |
> | LoFiT | 8h11min| 65GB |
> | RestoreLCC | 4h13min | 65GB |
>
> (2) Theoretical analysis. For RestoreLCC, the main overhead comes from: (i) trainable parameters (the scaling factor β in Eq. 8 and one bias term per head), (ii) intermediate activations (components vᵢ in Eq. 8), and (iii) coding efficiency due to modifications in the attention head output.
>
> (i) RestoreLCC introduces **fewer trainable parameters** than PEFT methods such as LoRA and DoRA (see Section 4.3 and Appen. C for details)
>
> (ii) **The overhead from intermediate activations (vᵢ) is extremely small and can be ignored**. The activations are fixed (frozen) during training. Each attention head has 128 dimensions; by the property of SVD this corresponds to at most 128 components, i.e., 128 × 128 = 16,384 parameters per head. Even in the extreme case of using all 1,024 heads, the adds only 1,024 × 16,384 ≈ 16.8M parameters - only about 0.24% of a 7B model. Moreover, as shown in Section 3 and Figure 8, only 10–30% of heads are required for recovery, so the actual overhead is much smaller.
>
> (3) **We optimized our code implementation to avoid inefficient loops**. Unlike LoFiT, which iterates over each head in thier code, our method uses only matrix multiplications to update all heads. This eliminates costly looping inside the Transformer. Consequently, RestoreLCC trains much faster than LoFiT; re‑implementing LoFiT with our approach similarly reduces its training time, confirming that the efficiency gain comes from our design.
>
> **In summary, RestoreLCC introduces smaller overhead than advanced PEFT methods.**
>
> The core code implementation is as follows:
> ```
> # (B:batch_size, T: sequence_length
> # H:head number per layer, D: head_dimension)
> out = attn_output.view(B, T, H, D)
> # comp: [H, D, D]:decomposed fixed components for heads
> # beta [H,D]: beta in Eq.8
> # b [H,D]: bias in Eq.8
> added_info = (
>     torch.bmm(comp.transpose(1, 2),  # (H, D, D)
>               beta.unsqueeze(2))  # (H, D, 1)
>     .squeeze(2)
> )
> b = b.view(1, 1, H, D)
> add = added_info.view(1, 1, H, D)
> # modify every head in one go
> attn_output = out  + add + b
> attn_output = attn_output.view(B, T, H*D)
> ```
> Notably, only a subset of attention heads use non‑zero values for comp (frozen, no gradients), beta (trainable), and b (trainable). GPUs will automatically optimize both the storage and the arithmetic for these zero‑valued vectors.
>
> [**W3**] Overhead sensitivity to number of parameters.
>
> [**R3**] We would like to state that our method is **not sensitive to the number of trainable parameters**.
>
> (1) First, as in our response [**R2**] to your [**W2**], RestoreLCC introduces less overhead than other advanced PEFT methods.
>
> (2) Empirical results. The number of trainable parameters in RestoreLCC depends on the proportion of attention heads selected for tuning. The table below reports GPU memory and training time for different ratios of heads. The results show: (i) The GPU memory consumption remains almost unchanged as trainable parameters increase and constantly lower than DoRA (71 GB). (ii) The training time is mostly smaller than DoRA and LoFiT.
>
> | head ratio| 0.1| 0.2| 0.3| 0.5|1|
> |----|-----|-----|--|-------|-----|
> |GPU| 65GB |65GB| 65GB|66GB|68GB|
> |time| 3h40min | 4h13min | 4h40min | 5h30min | 7h42min |
>
> (3) We would like to highlight that more parameters do not necessarily lead to better performance and thus do not introduce sensitivity. As shown in Section 3, not all attention heads are equally useful for recovery. Also, figure 8 demonstrates that using only 10%–30% of the most discriminative heads is sufficient to achieve the best performance. This is also why our method first contrastively probes important heads.
>
> [**Q1**] Results explanation on Figure 1.
>
> [**R4**] Thank you for raising this point. The improvement are for two reasons:
>
> (1) Task‑specific information. The added components act as task‑specific vectors (often referred to as task function vectors [6] or task steering vectors [7]). Consistent with these prior studies, such vectors provide task‑aware guidance and significantly enhance the model’s performance over the original  model.
>
> (2) Removal of useless or harmful components.Some lost components in each attention head can even be harmful, as they may introduce noise that negatively affects the model’s output. By recovering only the useful components and excluding the harmful ones, the model’s performance improves.
>
> [6] Function Vectors in Large Language Models, ICLR 2024
>
> [7] Semantics-Adaptive Activation Intervention for LLMs via Dynamic Steering Vectors, ICLR 2025
>
> [**Q2**] Comparison with full-parameter tuning.
>
> [**R5**] We conduct experiments on LLaMA‑7B. Code implementation is from Wanda. The average results are as follows. On Wanda‑pruned models, RestoreLCC achieves slightly better performance than FT. On SparseGPT‑pruned models, RestoreLCC is slightly worse than FT. Overall, RestoreLCC achieves comparable to FT.
>
> |      | ave. |    | ave. |
> |-----|---|-----|-----|
> | Wanda | 54.09| SparseGPT | 48.99   |
> | LoFiT | 56.82 | LoFiT| 52.11   |
> | FT | 58.43 | FT | 56.44 |
> | Ours  | 58.83 | Ours | 55|
>
> [**Q3**] Effects of using FFNs.
>
> [**R6**] We apply RestoreLCC to FFNs and evaluated different numbers of FFN layers. The best result obtained was 57.00, which is lower than the 58.83 achieved when using attention heads.
>
> | FFN ratio | 0.1 | 0.3 |0.5 | 0.7 |1 |
> |-----|-----|-----|-------|-------|------|
> | ave | 55.92 | 57  | 56.24 | 56.68 | 56.8 |
>
> [**Q4**] Details for PEFT implementations.
>
> [**R7**] We apologize for not providing these details. For LoRA and DoRA, we use the same settings: α = 16 and rank = 8. Regarding the applied modules, we try two configurations: (1) ["v_proj", "o_proj"], which tunes only the head output matrices; and (2)["q_proj", "k_proj", "v_proj", "o_proj"]`, which tunes all head matrices. The best results are reported in the paper.
>
> For LoFiT, we experiment with 10%, 20%, and 30% of the heads and report the best results.
>
> The batch size is set to 8. The learning rate is 1e-4. The max sequence length is 512. The number of training epochs is 2 for global recovery and 5 for task-specific recovery. All experiments are conducted on a single H100 GPU.
>
> **Thank you again for taking the time to review our paper. We will incorporate these discussions into the revised version and look forward to your feedback.**

---

> > ### Comment · Reviewer_Kk22 · 2025-08-06
> >
> > Thank you for your response! Your reply has partially addressed my concerns. Based on your answer, it seems that RestoreLCC is primarily designed to compensate for the information loss caused by pruning in the Attention module, rather than the knowledge loss resulting from pruning in the FFN module. Is my understanding correct?

---

> > > ### Author Response · Authors · 2025-08-06
> > >
> > > Dear Reviewer Kk22,
> > >
> > > Thank you for taking the time to review our responses and for your follow-up question.
> > >
> > > Yes, your understanding is correct. For a pruned model in which both attention heads and FFNs have been pruned, we identify only a few important attention heads and compensate for their pruning, without relying on the FFNs. **Our method can be directly applied to FFNs but we still choose attention heads for restoration. This is because restoring these key attention heads alone is sufficient to effectively recover the performance of the entire pruned model**. To further support this point, we offer the following perspectives:
> > >
> > > (1) Studies on mechanistic interpretability show that **attention heads specialize in distinct functions for different tasks, while FFNs mainly store knowledge and map inputs to outputs**. For example, [1] finds that in arithmetic tasks, attention heads process key information and FFNs then produce the final answer. They also show that fine‑tuning only important heads outperforms tuning all parameters (including FFNs). Similarly, [2] demonstrates that different heads play distinct roles, and [3] also supports this observation. By contrast, FFNs store factual knowledge [4] and refine output logits [5]. Hence, compensating attention heads is more appropriate than compensating FFNs.
> > >
> > > (2) Attention heads provide finer information, whereas FFNs produce high‑dimensional, aggregated outputs. In LLaMA-7B, each head outputs 128 dimensions (32 heads per layer), while an FFN outputs 4096. Smaller head outputs allow finer analysis. Moreover, choosing 32 heads spans multiple layers, while the same size in FFN covers only one layer.
> > >
> > > (3) Empirical evidence. It is worth mentioning that **our method can be directly applied to FFN modules**. Therefore, we applied RestoredLCC to FFNs and report the best results as follows. Compensating attention achieves 58.83, while compensating FFNs only reaches 57.00.
> > >
> > > |                   | BoolQ | RTE   | Hell. | Wino. | ARC-e | ARC-c | OBQA | Mean  |
> > > |-------------------|-------|-------|-----------|------------|-------|-------|------|-------|
> > > | Recover attn. | 72.84 | 69.68 | 56.34     | 65.98      | 71.8  | 40.96 | 34.2 | 58.83 |
> > > | Recover FFNs      | 69.42 | 63.54 | 55.54     | 67.17      | 70.96 | 39.59 | 32.8 | 57    |
> > >
> > >
> > > The results for different numbers of FFN layers are as follows:
> > >
> > > | FFN ratio | 0.1   | 0.3 | 0.5   | 0.7   | 1    |
> > > |-----------|-------|-----|-------|-------|------|
> > > | ave       | 55.92 | 57  | 56.24 | 56.68 | 56.8 |
> > >
> > > **In summary, considering both efficiency and effectiveness, we opt to restore attention modules rather than FFNs when recovering pruned models. Nonetheless, we acknowledge that combining attention heads with FFNs might potentially yield even better performance. To the best of our knowledge, this is the first study to explore restoring only a few crucial components to recover an entire pruned model. Therefore, investigating the combination of attention heads and FFNs is highlighted as part of our future work.**
> > >
> > >
> > > [1] Interpreting and improving large language models in arithmetic calculation, ICML 2024
> > >
> > > [2] Model Tells You What to Discard: Adaptive KV Cache Compression for LLMs, ICLR 2025
> > >
> > > [3] Dressing up llm: Efficient stylized question-answering via style subspace editing. ICLR 2025
> > >
> > > [4] Locating and Editing Factual Associations in GPT, NeurIPS 2022
> > >
> > > [5] Transformer Feed-Forward Layers Are Key-Value Memories, EMNLP 2021.

---

> > > > ### Comment · Reviewer_Kk22 · 2025-08-06
> > > >
> > > > Thank you for your response. I will raise my score to 4.

---

> > > > > ### Author Response · Authors · 2025-08-06
> > > > >
> > > > > Dear Reviewer Kk22,
> > > > >
> > > > > Thank you very much for your prompt response. We appreciate your comments and will incorporate all the discussed points into the revised paper.
> > > > >
> > > > >
> > > > > Best regards,
> > > > >
> > > > > Authors of Paper 7969

---

### Decision · Program_Chairs · 2025-09-17

**Decision:**

Accept (spotlight)

**Comment:**

(a) Summary

This paper introduces RestoreLCC, a novel and targeted restoration strategy for pruned large language models (LLMs). Unlike prior parameter-efficient fine-tuning (PEFT) methods that overlook the unique properties of pruned models, RestoreLCC explicitly identifies and compensates for the lost information caused by pruning. The authors perform an empirical study to gain insight into how leveraging the information in lost activations can be used to restore quality. Using these insights, RestoreLCC injects the lost information back into the model via a learned bias term in a three step process: (1) contrastive probing identifies the important attention heads, (2) Lost Component Compensation (LCC) performs SVD on the difference between the outputs of the important dense and pruned heads and learns optimal magnitudes for the top components before (3) augmenting the important heads with these learned bias terms, "re-injecting" the lost information back into the model. Empirical studies show improvements on most metrics compared to baselines.
Extensive experiments across multiple pruning schemes (structured, semi-structured, unstructured) and LLM variants demonstrate that RestoreLCC achieves superior accuracy and perplexity recovery without compromising sparsity or inference efficiency.

(b) Strengths

The paper introduces a unique insight—model degradation from pruning is traceable to lost activation components—which leads to a targeted and explainable restoration method. The work addresses a critical bottleneck in deploying sparse LLMs by offering a low-overhead, generalizable recovery mechanism. It shows consistent gains over strong PEFT baselines like LoRA, DoRA, and LoFiT.

The paper observes that pruning-induced information loss is reflected in attention activations, and that selectively restoring key components can significantly recover model performance.

The methodology is well-justified with theoretical and empirical analyses. Ablation studies, probing diagnostics, and interpretability results are thorough and convincing. The method is validated under different pruning settings.

The overall exposition is clear, well-structured, and supported with informative figures (e.g., logit gain visualizations, component effects).

The solution's low-overhead is particularly compelling; additional bias terms are a very small tax on model size and deployment efficiency, in contrast to the LoRA family of techniques, which require additional (though relatively small) weight matrices. The contrastive probing and compensation process is adequately described such that an expert could reproduce the results. The evaluations are sufficiently broad with respect to models and tasks to inspire confidence that there are no surprises hidden behind under-explored areas. This particular solution to the problem incorporates existing ideas, but is largely original.

(c) Weaknesses

All compared PEFT methods are limited to very small trainable parameter settings. I believe this setup is favorable to the method. However, it is clear that the training overhead of PEFT is not sensitive to the number of trainable parameters; for example, the overhead difference between rank=1 and rank=16 is minimal. I think that in the recovery fine-tuning stage, performance recovery should be the main focus, and a higher rank may yield better performance with only negligible increases in fine-tuning overhead.

The contrastive probing module depends on data/model-specific classifier training, which could affect practical deployment and cross-task portability.

Clarity is a mixed bag - while the exposition is clear, some details are not. For example, the precise sparsity imposed is unclear. At times the activations are pruned (line 113), but later it is the attention heads (I assume weights) that are pruned (line 132). It's also not clear which weights are pruned; is it only those in the attention blocks Q, K, V and dense output projections, or also weights in the MLP blocks (which can be much larger due to the expansion factor)? This has direct bearing on the significance of the submission. Also, consider avoiding the term "performance;" it is overloaded and can mean both model quality as well as execution speed or efficiency. In this context, where both meanings are plausible, it's better just to be explicit.

(d) Most important reasons for decision to accept/reject.

The novel and effective approach to restoring performance after network pruning, and the admirable job the authors did in responding to the reviewers' critiques.

(e) Summary of the discussion and changes during the rebuttal period.

The authors provided extensive and convincing rebuttal text, with additional experiments as requested by the reviewers. All reviewers raised their scores after the rebuttal period.

Reviewer n2VC stated that the authors provided an exceptionally strong and thorough rebuttal that has fully addressed all of their initial concerns, with no remaining unresolved issues. The new experiments and clarifications have significantly strengthened the paper's claims and overall contribution.
The authors added extensive experiments on entirely different and recent architectures (Qwen3, DeepSeek-R1-Qwen3), demonstrating that RestoreLCC consistently outperforms baselines. This provides compelling evidence for the method's broader applicability.

Questions about the practical deployment, specifically the portability of the probing module and the method's efficiency at scale, were also fully addressed with new, targeted experiments. The authors demonstrated tsuccessful cross-task performance recovery without retraining the probing module, directly answering the scalability question, and showed effective performance on a much larger 70B model (LLaMA-70B), proving scalability. A concrete latency benchmark was provided showing a negligible inference overhead of ~3%, confirming real-world efficiency.

The weight of these resolved issues is high, as they were central to validating the paper's core claims of being a generalizable and practical method. The authors' diligence in running these new experiments has transformed the paper from a promising one with limitations to a robust and well-supported contribution.

Reviewer n2VC stated "Your rebuttal was exemplary. The new experiments were precisely what was needed to validate your claims and were highly persuasive. The paper is now substantially stronger, with robust evidence backing the generalizability, cross-task portability, and efficiency of RestoreLCC."